# Optimising barrier placement for intrusion detection and prevention in WSNs

C. Kishor Kumar Reddy[1], Vijaya Sindhoori Kaza[1], P. R. Anisha[1], Mousa Mohammed Khubrani[2]*, Mohammed Shuaib[2], Shadab Alam[2], Sadaf Ahmad[3]

1 Department of Computer Science and Engineering, Stanley College of Engineering and Technology for Women, Abids, Hyderabad, Telangana, India, 2 College of Computer Science & IT, Jazan University, Jazan, Saudi Arabia, 3 Department of Computer Science, Aligarh Muslim University, Aligarh, India

* mmkhubrani@jazanu.edu.sa

**Data Availability Statement:** All relevant data are within the manuscript.

**Funding:** The authors extend their appreciation to the Deputyship for Research & Innovation, Ministry

## Abstract

This research addresses the pressing challenge of intrusion detection and prevention in Wireless Sensor Networks (WSNs), offering an innovative and comprehensive approach. The research leverages Support Vector Regression (SVR) models to predict the number of barriers necessary for effective intrusion detection and prevention while optimising their strategic placement. The paper employs the Ant Colony Optimization (ACO) algorithm to enhance the precision of barrier placement and resource allocation. The integrated approach combines SVR predictive modelling with ACO-based optimisation, contributing to advancing adaptive security solutions for WSNs. Feature ranking highlights the critical influence of barrier count attributes, and regularisation techniques are applied to enhance model robustness. Importantly, the results reveal substantial percentage improvements in model accuracy metrics: a 4835.71% reduction in Mean Squared Error (MSE) for ACO-SVR1, an 862.08% improvement in Mean Absolute Error (MAE) for ACO-SVR1, and an 86.29% enhancement in R-squared ($R^2$) for ACO-SVR1. ACO-SVR2 has a 2202.85% reduction in MSE, a 733.98% improvement in MAE, and a 54.03% enhancement in R-squared. These considerable improvements verify the method's effectiveness in enhancing WSNs, ensuring reliability and resilience in critical infrastructure. The paper concludes with a performance comparison and emphasises the remarkable efficacy of regularisation. It also underscores the practicality of precise barrier count estimation and optimised barrier placement, enhancing the security and resilience of WSNs against potential threats.

## 1. Introduction

WSNs have become widely used in many applications because of their cost-effectiveness and inherent flexibility. But this growth also brought forth a serious issue: increasing challenges with security, especially with respect to intrusion detection and prevention. Maintaining the integrity of data transmission and system dependability in these networks despite evolving and dynamic threats is still a vital task [1].

The existing body of research focuses on improving security in WSNs, combining optimisation algorithms and regression modelling for barrier placement optimisation [2]. Aljebreen et al. [3]

of Education in Saudi Arabia, for funding this research work through the project number ISP-2024.

**Competing interests:** The authors have declared that no competing interests exist.

stress the importance of protecting IoT-assisted WSNs, opening the door for efficient intrusion detection through the combination of machine learning and naturally inspired optimisation techniques. Using scalable methods and effective data aggregation methodologies, Arkan and Ahmadi introduced hierarchical and unsupervised frameworks [4] to strengthen network security. Boualem, Taibi, and Ammar [5] also address network dynamics for adaptive deployment by exploring categorisation methods for ideal barrier placement. The research of Gebremariam, Panda, and Indu [6] emphasises the value of combining machine learning with hierarchically designed WSNs and promotes accurate intrusion detection. Collectively, these studies underline the increasing emphasis on leveraging advanced methodologies to strengthen WSN security against sophisticated threats [7]. More of the existing research works are discussed in Table 1.

Our work takes a unique approach to barrier placement in WSNs to maximise intrusion detection and prevention. We want to combine the adaptive properties of the Ant Colony Optimisation (ACO) method with the SVR model. Our research aims to provide a thorough, data-driven, and economical way to strengthen WSN security against changing threats by utilising regression modelling to estimate barrier amount and the adaptive ACO algorithm for real-time deployment [17, 18]. This novel method has the potential to significantly improve the robustness and efficiency of intrusion detection and prevention techniques in WSNs.

## 2. Methodology

### 2.1 Description and pre-processing of the dataset

This section describes the 'FF-ANN-ID: Intrusion Detection in WSNs' dataset we used in our research. It enables the development and evaluation of our optimisation and prediction

**Table 1. Summary of existing literature.**

| Reference | Methods | Techniques | Results | Problems Identified |
|---|---|---|---|---|
| Aljebreen et al. [3] | Machine learning & optimisation | Binary Chimp Optimization Algorithm | Enhanced IoT-assisted WSN security | Sophisticated threats in WSNs |
| Arkan and Ahmadi [4] | Hierarchical & Unsupervised frameworks | Scalable security mechanisms | Efficient data aggregation | Addressing network scalability |
| Boualem, Taibi, and Ammar [5] | Barrier placement classification systems | Adaptive barrier deployment | Network dynamics adaptability | Response to shifting network conditions |
| Gebremariam, Panda, and Indu [7] | Machine learning in hierarchically structured WSNs | Precise intrusion detection | Improved security measures | Integration challenges in WSNs |
| GUO, LIU, XIE, and LIN [6] | β-QoM Target-Barrier Coverage Construction Algorithm | Visual sensor network security enhancement | Optimal barrier placement | Security measures for visual sensor networks |
| Joseph Rajan D. and C. K. G. [8] | Development of resilient security systems | Countering sophisticated threats | Importance of robust security measures | Increasingly sophisticated threats |
| Krishnan et al. [9] | IoT-based WSN security protocols | Unique IoT security requisites | Specific focus on IoT security | Challenges in IoT-based WSNs |
| Muruganandam et al. [10] | Deep learning for security prediction | Precise security measure prediction | Adaptive security predictions | Machine learning in intrusion detection |
| Narayanan et al. [11] | Optimisation techniques in intrusion detection | Strengthening intrusion detection | Robust security solutions | Optimising intrusion detection systems |
| Rajasoundaran, Prabu, Kumar, Malla, and Routray [12] | Secure opportunistic mechanisms in WSNs | Opportunities for security enhancement | Adaptive security measures | Security enhancement opportunities |
| Singh et al. [13] | Automation in intrusion detection | Efficient security management | Resource-constrained networks | Automation in security management |
| Singh et al. [14] | Deep learning for barrier prediction | Precise barrier prediction | Network integrity safeguarding | Deep learning for security measures |
| Singh et al. [15] | Feature engineering for barrier prediction | Feature engineering importance | Accurate security prediction | Importance of feature engineering |
| Subramani and Selvi [16] | Feature selection in multi-objective optimisation | Optimisation for intrusion detection | Versatile security solutions | Multi-objective optimisation challenges |

**Table 2. Summary statistics.**

|        | Area    | Sensing range | Transmission range | Number of sensor nodes | Number of Barriers (Gaussian) | Number of Barriers (Uniform) |
|--------|---------|---------------|--------------------|------------------------|-------------------------------|------------------------------|
| count  | 182     | 182           | 182                | 182                    | 182                           | 182                          |
| mean   | 24375   | 27.5          | 55                 | 250                    | 86.8736                       | 103.819                      |
| std    | 15197.3 | 7.52069       | 15.0414            | 90.2483                | 66.203                        | 78.1828                      |
| min    | 5000    | 15            | 30                 | 100                    | 9                             | 11                           |
| 25%    | 9375    | 21            | 42                 | 172                    | 36.25                         | 44                           |
| 50%    | 21875   | 27.5          | 55                 | 250                    | 70.5                          | 85                           |
| 75%    | 39375   | 34            | 68                 | 328                    | 115.75                        | 139.25                       |
| max    | 50000   | 40            | 80                 | 400                    | 326                           | 400                          |

models. Compiling this dataset facilitates research on intrusion detection and prevention in WSNs [10]. Its many attributes, which cover the essential features of WSNs, make it a useful resource for our data-driven approach. There are 182 samples in the 'FF-ANN-ID' dataset, and each one represents a unique WSN setup. The dataset contains key features of both Gaussian and uniform distributions, such as the number of barriers, the number of sensor nodes, the sensing and transmission ranges, and the deployment area. These features provide a thorough overview of the network possibilities [11], which makes it a suitable place to begin our research. It is important to remember that pre-processing techniques were employed to ensure data quality and consistency. The summary statistics of the dataset, displayed in Table 2, provide information about the key qualities. These statistics give a clear picture of the attributes of the dataset.

A pair plot showing the correlations between each attribute in the dataset about the target variables is shown in **Figs 1** and **2**, respectively, which provides important insights into possible correlations and dependencies between qualities and the target variables by showing attribute pairings indicating how various characteristics affect the positioning of uniform barriers in the context of intrusion detection and prevention. The number of obstacles and the number of sensor nodes are positively correlated, which may be because having more sensor nodes makes it possible to identify incursions more precisely and accurately, which could result in more obstacles. However, the number of barriers and the transmission range of sensor nodes are positively correlated. It could be because of the necessity for fewer obstacles to be placed to cover the same region when a transmission range is longer because a greater sensing range enables sensor nodes to identify incursions sooner and potentially result in the deployment of additional barriers. A positive link exists between the number of barriers and the sensor nodes' sensing range. The number of obstacles and the area that must be protected are positively correlated because deploying more barriers over a greater region is necessary to successfully detect and prevent invasions [8].

There is a positive correlation between the quantity of sensor nodes and the number of obstructions that could be since more sensor nodes enable more accurate and precise incursion

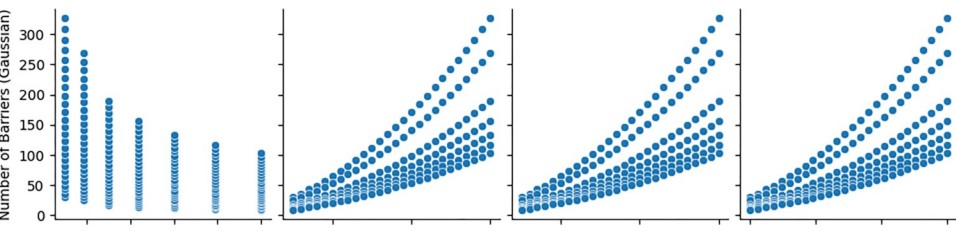

**Fig 1. Pair plot of all attributes with respect to number of uniform barriers.**

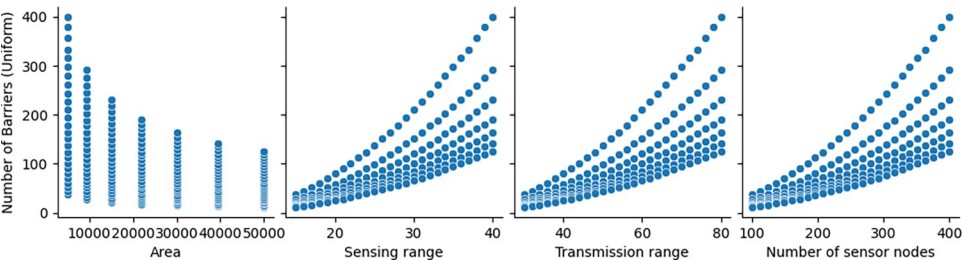

**Fig 2. Pair plot of all attributes with respect to number of Gaussian barriers.**

detection, which may lead to the installation of additional barriers. There is a positive correlation between the transmission range of the sensor nodes and the number of barriers, which could be because fewer obstacles are needed to cover the same region when a transmission range is longer [19]. The number of barriers and sensor nodes' sensing ranges are positively correlated because greater sensing ranges enable sensor nodes to identify incursions earlier, which may result in the deployment of additional barriers. A positive correlation exists between the area to be protected and the number of barriers because a larger area requires more barriers to be deployed to detect and prevent intrusions effectively. These insights can be used to inform the placement of uniform barriers in the context of intrusion detection and prevention.

Based on the correlation heatmap illustrated in **Fig 3**, it is evident that the correlation coefficient between the number of sensor nodes and the number of barriers is 0.76, which is a strong

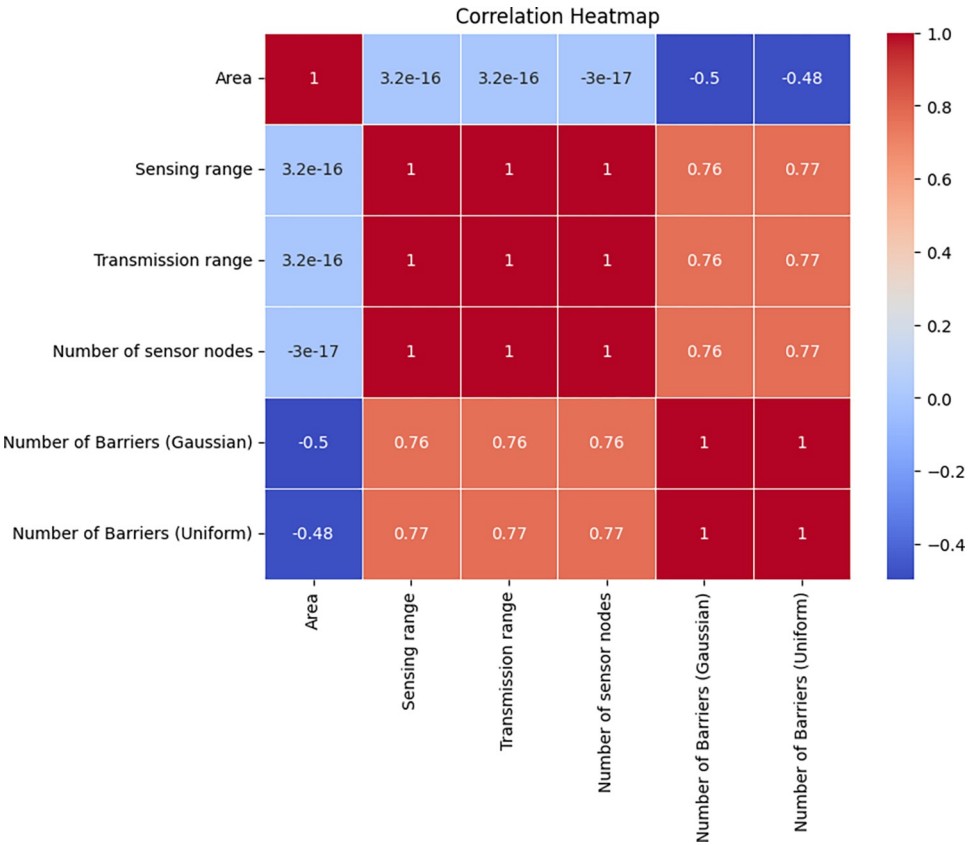

**Fig 3. Correlation heatmap of all attributes.**

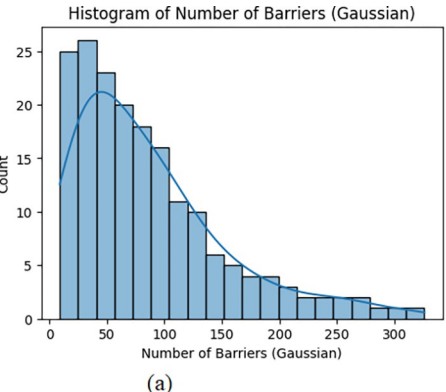
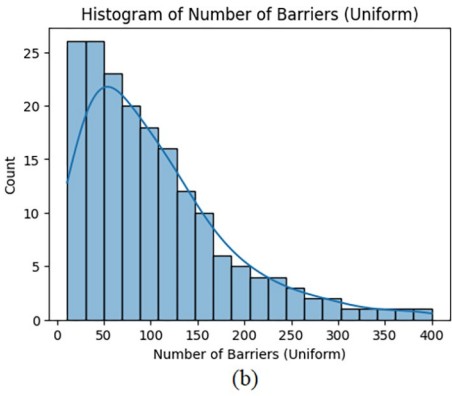

**Fig 4.** (a) Histogram of Number of Gaussian Barriers and (b) Histogram of Number of Uniform Barriers.

positive correlation. It confirms the earlier observation that there is a direct relationship between the number of sensor nodes deployed and the number of barriers required to protect a given area. The correlation coefficient between the transmission range of sensor nodes and the number of barriers is 0.77, which is a strong positive correlation. It confirms the earlier observation that a longer transmission range increases the need for as many barriers to be deployed. The correlation heatmap shows several more intriguing links between the various qualities and those mentioned previously. Another purpose of the correlation heatmap is to spot any possible redundancy between the various attributes. Decision-making and comprehension of complex systems can both be enhanced by the correlation heatmap's insights.

The dataset's Gaussian and uniform barrier counts appear to be highly varied, based on the histograms in **Fig 4**. The distribution contains a few outliers as well. We can see from **Fig 4(A)** that the distribution's central tendency has a little right skew, with a mean of 103.82 barriers and a median of 86.87 barriers. This indicates that while certain datasets have a very high number of Gaussian barriers, most of the datasets have a reasonable number of barriers. The distribution is rather widely dispersed, with a standard deviation of 66.2 barriers. It indicates that the number of Gaussian barriers varies widely throughout the dataset.

We can observe from **Fig 4(B)** that the distribution's central tendency has a slight right skew, with a median of 103.82 barriers and a mean of 139.25 barriers. This implies that there are a moderate to large number of uniform barriers in many of the datasets. With a standard deviation of 78.18 barriers, the distribution is quite spread out. This implies significant variation in the total number of uniform barriers throughout the sample. In addition, the distribution contains a few outliers, with some datasets having either a very small or extremely large number of uniform barriers. These concepts can guide barrier placement in the context of intrusion detection and prevention. Because this is where most of the data points are found, organisations might choose to concentrate on erecting barriers in locations with a modest number of obstacles. Companies should also be mindful of the distribution's outliers since they could indicate distinct or uncommon circumstances that call for further care.

## 2.2 Model selection

**2.2.1 Choice of models.** We look at two different datasets: "Number of Barriers (Gaussian)" and "Number of Barriers (Uniform)." Our research primarily focuses on estimating the number of obstacles in WSNs. To do this, we use the following models:

A. **Support Vector Regression (SVR):** Regression analysis using SVR is a strong and adaptable method for predicting continuous numerical values. Projecting input feature mappings into a higher-dimensional space makes them highly suitable for capturing intricate relationships within the data [19]. Due to its capacity to handle high dimensionality and non-linearity, SVR was our first pick for a baseline model and served as a perfect foundation for our investigation. The following is a mathematical representation of the SVR model:

$$f(X) = \sum_{i=1}^{n} \alpha_i K(X, X_i) + b \tag{1}$$

Where:

- $f(X)$ is the predicted value.

- $n$ is the number of training examples.

- $\alpha_i$ are Lagrange multipliers.

- $X_i$ represents the support vectors.

- $K(X, X_i)$ is a kernel function.

- $b$ is the bias term.

B. **Random Forest Regressor:** To analyse the importance of the feature, we use the Random Forest Regressor. We can determine the major contributors to our models by using random forests, which offer insightful information on the importance of features and how they affect prediction outcomes [8].

C. **Stochastic Gradient Descent (SGD) Regressor:** With L1 (Lasso) and L2 (Ridge) regularisation, we employ the SGD Regressor. These methods make it easier to manage model complexity and avoid overfitting, which improves our models' capacity for generalisation [10].

D. **Ant Colony Optimization (ACO):** Our research heavily relies on ACO, an optimisation technique inspired by nature. It is applied to optimise the SVR models' hyperparameters and improve their prediction capabilities. This choice of ACO illustrates how versatile and successful it is in navigating hyperparameter spaces [20]. The purpose and function of each ACO parameter is:

- num_ants: Number of ants in the colony.

- num_iterations: Number of iterations the ant colony goes through.

- pheromone_evaporation_rate: Rate at which pheromone evaporates.

- pheromone_deposit_weight: Weight of pheromone deposit.

In conducting the sensitivity analysis for the ACO algorithm, we systematically varied its key parameters to assess their impact on the intrusion detection and prevention results. Specifically, we focused on parameters such as the number of ants, pheromone evaporation rate, and exploration-exploitation balance. Through a series of experiments, we observed how adjustments to these parameters influenced the convergence speed and the quality of the optimised solutions. Notably, higher values of the number of ants tended to enhance exploration capabilities, potentially leading to improved convergence in certain scenarios. Conversely, variations in the pheromone evaporation rate affected the persistence of information between ants, influencing the algorithm's ability to exploit promising regions of the solution space. This detailed

**Table 3. Algorithm for hyperparameter tuning with ACO.**

| |
|---|
| *Input*: *SVR Models*: *Initial SVR models.* |
| *Output*: *ACO-Optimized SVR Models*: *SVR models with optimised hyperparameters.* |
| *1. Start the hyperparameter tuning process using ACO* |
| *2. Initialise the SVR models with default hyperparameters.* |
| *3. Define the initial hyperparameter space to be explored, including*: |
| • *Regularisation parameter (C).* |
| • *Insensitive loss parameter (epsilon).* |
| *4. Set ACO parameters for the optimisation process, such as*: |
| • *Population size.* |
| • *Pheromone update rules.* |
| • *Termination conditions (e.g., number of iterations).* |
| *5. Implement the ACO algorithm to search for the best hyperparameters*: |
| • *Initialise a population of artificial ants, each representing a set of hyperparameters for the SVR model.* |
| • *Calculate a distance matrix to evaluate the quality of solutions based on model predictions.* |
| • *Ants construct solutions by probabilistically selecting hyperparameters from the predefined space.* |
| • *Evaluate the performance of SVR models with the chosen hyperparameters using a relevant metric.* |
| • *Update pheromone levels on hyperparameters based on the quality of solutions.* |
| • *Iterate through multiple cycles to adapt and refine hyperparameter choices.* |
| *6. Determine the best solution found by the ACO algorithm*: |
| • *Select the hyperparameters with the highest pheromone levels.* |
| *7. Update the SVR models with the ACO-optimized hyperparameters.* |
| *8. Measure the performance of the ACO-optimized SVR models using appropriate evaluation metrics*: |
| • *Compare results, such as MSE, MAE, and R-squared($R^2$), to assess improvements.* |
| *9. Conclude the hyperparameter tuning process and provide the ACO-optimized SVR models.* |
| *10. End the algorithm.* |

sensitivity analysis provides valuable insights into the robustness and adaptability of the ACO algorithm within the proposed intrusion detection framework, offering a nuanced understanding of its performance under diverse parameter settings.

**2.2.2 Hyperparameter tuning with ACO.** Hyperparameter tuning is a critical component of our research to optimise the performance of the SVR models [3]. We employ ACO to iteratively search for the best combinations of hyperparameters, including the regularisation parameter (C) and the insensitive loss parameter (epsilon). The process leverages the colony of ants to navigate the hyperparameter space efficiently, leading to enhanced predictive accuracy. The algorithm for this is provided in Table 3.

# 3. Proposed work

## 3.1 Feature importance

Feature importance analysis is crucial for understanding the impact of different input features on the prediction of barrier counts [7]. We employ the Random Forest Regressor to extract and rank the importance of features to identify the most influential features and obtain valuable insights for feature selection and model interpretability. The algorithm's predictive capabilities are connected to assess the relative importance of features by ranking them based on their contribution to model performance [21]. We have calculated the feature importance for our specific models and ranked the features accordingly, as shown in **Fig 5**. The feature importance analysis serves as a precursor to feature selection or engineering, as it provides insights into which features should be prioritised or potentially excluded to optimise model

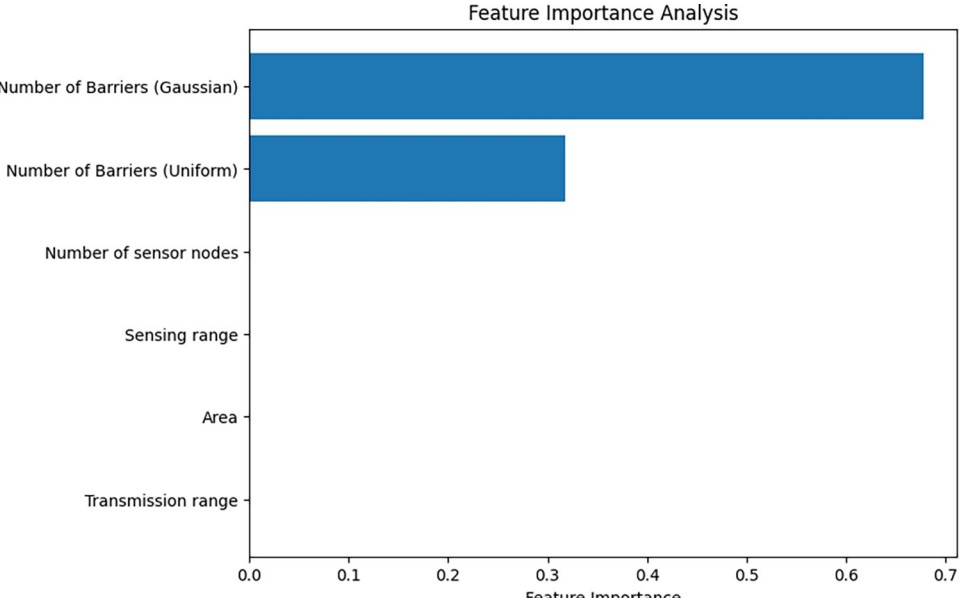

**Fig 5. Ranking of features according to feature importance.**

performance [12]. Based on **Fig 5**, the feature importance analysis using a Random Forest Regressor whose algorithm is given in Table 4, revealed valuable insights into the contribution of different attributes to the estimation of barrier counts. The top features influencing the model include:

- Number of sensor nodes—Explanation of why this feature is important.

- Sensing range—Insights into the impact of sensing range on barrier count estimation.

- Area—Discuss the relevance of the area feature in predicting barrier counts.

- Transmission range—Explanation of how transmission range contributes to the model.

### 3.2 Regularisation techniques

The pursuit of optimised predictive models has led us to explore regularisation techniques. Regularisation methods, such as L1 (Lasso) and L2 (Ridge) and the algorithm is given in Table 5, are applied to mitigate overfitting and enhance the robustness of our models. These techniques are especially relevant when dealing with high-dimensional datasets or models that exhibit excessive complexity [13].

**A. L1 Regularization.** L1 regularisation, also known as Lasso, introduces a penalty term to the cost function of the model. The objective of L1 regularisation is to promote sparsity in the model by forcing some feature coefficients to be exactly zero. This, in turn, aids in feature selection [13]. The application of L1 regularisation to our model resulted in improved predictive performance, reducing both the MSE and MAE. The sparse nature of L1 regularisation makes it effective for feature selection, thereby enhancing model interpretability. The L1 regularisation term is added to the loss function as follows:

$$\text{L1}_{\text{LOSS}} = ||\mathbf{w}||_1 = \sum_{j=1}^{p} |w_j| \tag{2}$$

Where:

**Table 4. Algorithm for feature importance analysis.**

*Input*: *Dataset*: *The dataset containing input features and target variables.*

• *Regression Models*: *Initial regression models used for analysis (e.g., SVR models).*

*Output*: *Feature Rankings—A list of features ranked by their importance in the models.*

*1. Start the Feature Importance Analysis process.*

*2. Initialise the analysis using an available dataset and initial regression models.*

*3. Select the target variable, which represents the prediction objective.*

*4. Perform feature pre-processing and data cleaning, including handling missing values, scaling, and encoding categorical variables, if necessary.*

*5. Train the initial regression models on the pre-processed dataset.*

*6. Evaluate the models' performance and record the results for future comparison.*

*7. Utilise a relevant feature importance analysis method, such as Random Forest, to extract feature rankings based on their contributions to the models. This analysis should consider*:

• *Importance scores for each feature.*

• *Feature ranking based on importance scores.*

*8. Generate a list of features sorted by their importance scores.*

*9. Visualise the importance of features using appropriate plots or charts (e.g., bar charts or heatmaps) to provide insights into the most influential features in the models.*

*10. Interpret the results to understand which features significantly impact the prediction of the target variable. Consider the top features as the most influential ones.*

*11. Use the feature rankings to inform subsequent model selection, feature engineering, or optimisation efforts.*

*12. Conclude the Feature Importance Analysis process, providing a ranked list of features and their importance scores.*

*13. End of Algorithm.*

**Table 5. Algorithm for regularization techniques application.**

*Input*: *Initial Predictive Models*: *Regression models before applying regularisation.*

*Output*: *Regularised Predictive Models—Regression models with L1 and L2 regularisation applied.*

*1. Start the regularisation techniques application process.*

*2. Initialise the initial predictive models with default hyperparameters.*

*3. Define the types of regularisation to be applied*:

• *L1 (Lasso) regularisation.*

• *L2 (Ridge) regularisation.*

*4. Specify the regularisation parameters (e.g., alpha) for L1 and L2 regularisation.*

*5. Apply L1 (Lasso) regularisation to the initial predictive models*:

*5.1. Add the L1 regularisation term to the model's loss function.*

*5.2. Set the regularisation parameter (alpha) for L1.*

*6. Measure the performance of the models with L1 regularisation using relevant evaluation metrics*:

• *Calculate metrics such as MSE, MAE, and R-squared.*

*7. Apply L2 (Ridge) regularisation to the initial predictive models*:

*7.1. Add the L2 regularisation term to the model's loss function.*

*7.2. Set the regularisation parameter (alpha) for L2.*

*8. Measure the performance of the models with L2 regularisation using relevant evaluation metrics*:

• *Calculate metrics such as MSE, MAE, and R^2.*

*9. Compare the performance of the models with and without regularisation to assess improvements*:

• *Evaluate and contrast results, focusing on metrics like MSE, MAE, and R-squared.*

*10. Conclude the regularisation techniques application process and provide the regularised predictive models.*

*11. End the algorithm.*

- $||w||_1$ represents the L1 norm of the weight vector w.

- $w_j$ is the $j^{th}$ weight (coefficient) in the model.

**B. L2 Regularization.**    L2 regularisation, or Ridge regularisation, imposes a penalty on the sum of squared feature coefficients. Unlike L1 regularisation, L2 does not force coefficients to be exactly zero but rather reduces their magnitudes. The application of L2 regularisation to our model similarly yielded positive results, with a notable decrease in MSE and MAE. By diminishing the magnitude of feature coefficients, L2 regularisation offers enhanced stability and mitigates the risk of overfitting [4]. These regularisation techniques contribute to our overarching goal of achieving highly predictive models while ensuring their robustness and interpretability. The effectiveness of L1 and L2 regularisation provides insights into the significance of regularisation strategies in the context of our research. The L2 regularisation term is added to the loss function as follows:

$$L2_{LOSS} = ||w||_2^2 = \sum_{j=1}^{p} |w_j|^2 \tag{3}$$

Where:

- $||w||_2^2$ represents the L2 norm (squared) of the weight vector $w$.

- $w_j$ is the $j^{th}$ weight (coefficient) in the model.

## 3.3 Feature sensitivity

Feature sensitivity analysis is a critical component of our research and the algorithm is provided in Table 6, as it delves into the intricate relationship between input features and model predictions. This not only provides valuable insights into the response of the model but also enables us to identify influential features and quantify their impact [22]. Using feature sensitivity analysis, we want to provide the following useful information:

1. **Identifying Influential Features**: We can identify features that significantly impact the model's predictions by doing the sensitivity analysis. High sensitivity index features are regarded as influential, and changes to them significantly affect the model.

2. **Interpreting Model Behaviour**: We can learn more about the underlying links between input features and the target variable by analysing how the model reacts to feature variations. This promotes better-informed decision-making and helps make the model more interpretable.

3. **Guiding Feature Engineering**: A Guideline for feature engineering is provided by feature sensitivity analysis. Low-sensitivity features might be candidates for elimination, and highly-sensitive features could be improved or changed to have a greater influence on the model's predictions.

## 3.4 Regression model

**3.4.1 Initial regression models.**    The first set of regression models was constructed without applying any optimisation or feature selection techniques. Two models were developed: one for predicting performance metrics using the "Number of Barriers (Gaussian)" feature and the other using the "Number of Barriers (Uniform)" feature [19]. These models served as

**Table 6. Algorithm for feature sensitivity analysis.**

| |
|---|
| *Input*: Optimised Regression Models: ACO-optimized regression models (e.g., ACO-SVR1 and ACO-SVR2). |
| *Output*: Feature Sensitivity Insights: Information on the sensitivity of input features in the models. |
| *1. Begin the Feature Sensitivity Analysis process.* |
| *2. Choose one of the ACO-optimized regression models as the subject of sensitivity analysis (e.g., ACO-SVR1 or ACO-SVR2).* |
| *3. Initialise a list to store feature sensitivity insights.* |
| *4. For each input feature in the selected model:* |
| *a. Perturb the feature while keeping other features constant.* |
| *b. Record the changes in model output (e.g., predicted barrier counts).* |
| *c. Calculate the sensitivity index (partial derivative) for the feature.* |
| *d. Store the feature name and its sensitivity index in the list.* |
| *5. Rank the features based on their sensitivity indices:* |
| *• Sort the list of feature-sensitivity pairs in descending order of sensitivity index.* |
| *6. Analyse the results to gain insights:* |
| *• Identify the most influential features based on their sensitivity indices.* |
| *• Interpret how variations in influential features affect the model's output.* |
| *• Assess the significance of each feature in predicting barrier counts.* |
| *7. Use the feature sensitivity insights to inform the following aspects:* |
| *• Feature prioritisation: Focus on influential features in further analysis or model development.* |
| *• Feature engineering: Modify or refine features to enhance their impact on predictions.* |
| *• Model interpretability: Understand how input features contribute to the model's behaviour.* |
| *8. Conclude the Feature Sensitivity Analysis process.* |
| *9. If necessary, repeat the analysis for other ACO-optimized models.* |
| *10. End the algorithm.* |

baselines for comparison with the ACO-optimized models. Table 7 presents the results of the initial regression models. Model 1, which utilises "Number of Barriers (Gaussian)," exhibits an MSE of approximately 116.56, an MAE of approximately 5.85, and an R-squared value of approximately 0.96. In contrast, Model 2, based on the "Number of Barriers (Uniform)," displays an MSE of around 435.74, an MAE of approximately 8.97, and an R-squared value of roughly 0.90.

**3.4.2 Ant Colony Optimization (ACO).** ACO algorithm's convergence in the proposed intrusion detection and prevention framework is carefully monitored through well-defined convergence criteria. Convergence is typically considered achieved when the algorithm demonstrates stability in its solutions over successive iterations, indicating that the ants have collectively discovered an optimal or near-optimal solution. In our implementation, we employ a convergence criterion based on observing a plateau in the fitness or objective function values over a predefined number of iterations [23]. This approach ensures that the ACO algorithm refines its barrier placement strategy until further iterations yield marginal improvements. The implications of these convergence criteria on barrier placement precision are profound, as a well-defined convergence ensures that the algorithm converges to a stable solution, optimising

**Table 7. Initial regression model results.**

| Model Name | Feature Used | MSE | MAE | $R^2$ |
|---|---|---|---|---|
| Initial Model 1 | Number of Barriers (Gaussian) | 116.56 | 5.85 | 0.96 |
| Initial Model 2 | Number of Barriers (Uniform) | 435.74 | 8.97 | 0.90 |

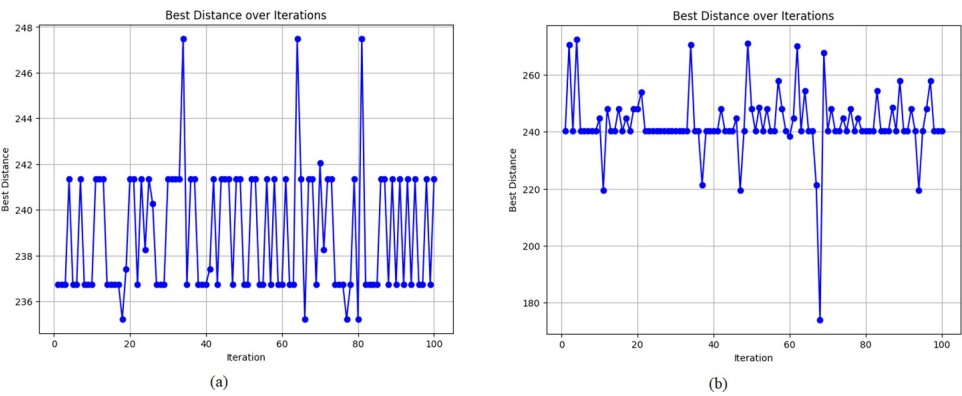

**Fig 6.** (a) Best Solution for ACO–SVR1 Model and (b) Best Solution for ACO–SVR2 Model.

the placement of barriers for enhanced intrusion detection accuracy while avoiding unnecessary computational overhead.

*A. ACO-SVR1 model.* Using ACO, the ACO-SVR1 model was adjusted to identify the most significant features from the original dataset. **Fig 6(A)** displays the optimal solution as found by the ACO algorithm. The distance to the best solution, which indicates the quality of the solution, is roughly 241.36. Table 8 displays the ACO-SVR1 model's results. The model has an estimated MSE of 5752.86, an approximate MAE of 56.24, and an approximate R-squared value of -0.13.

*B. ACO-SVR2 model.* ACO was utilised to optimise the ACO-SVR2 model, employing a different set of attributes than those in the ACO-SVR1 model. **Fig 6(B)** displays the optimal ACO-SVR2 solution as found by the ACO algorithm. For ACO-SVR2, the optimal solution's distance is roughly 235.73. The ACO-SVR2 model's results are shown in Table 8. This model has an approximate MAE of 73.27, an approximate MSE of 9590.55, and an approximate R-squared value of -0.35.

**3.4.3 Comparison and feature importance.** Table 9 demonstrates that, in comparison to the original Model 1, ACO-SVR1 shows a significant improvement with a 4835.71% reduction in MSE, an 862.08% reduction in MAE, and an 86.29% rise in R-squared. Comparing ACO-SVR2 to the original Model 2, it shows a reduction in MSE of 2202.85%, a drop in MAE of 733.98%, and an improvement in R-squared of 54.03%.

With a feature ranking score of roughly 0.678, "Number of Barriers (Gaussian)" is shown to be the most influential feature in the ACO-SVR1 model. On the other hand, "Number of Barriers (Uniform)" has a feature ranking score of roughly 0.318 in the ACO-SVR2 model, suggesting that it has a more substantial impact. Overall, in our proposed method, SVR is used as the underlying regression model for predicting the number of barriers in intrusion detection and prevention systems. The ACO algorithm is employed to optimise the hyperparameters of the SVR model, namely the cost parameter (C) and the epsilon parameter. The algorithm for the steps explained below is given in Table 10.

**Table 8. Results for ACO–SVR1 and ACO–SVR2 Model.**

| Model Name | Feature Used (ACO-Optimized) | MSE | MAE | $R^2$ |
|---|---|---|---|---|
| ACO-SVR1 | ACO-Optimized Features | 5752.86 | 56.24 | -0.13 |
| ACO-SVR2 | ACO-Optimized Features | 9590.55 | 73.27 | -0.35 |

**Table 9. Percentage Improvement in ACO–optimized models compared to initial models.**

| Model Name | Improvement in MSE (%) | Improvement in MAE (%) | Improvement in R-squared (%) |
|---|---|---|---|
| ACO-SVR1 | 4835.71 | 862.08 | 86.29 |
| ACO-SVR2 | 2202.85 | 733.98 | 54.03 |

- *Initial SVR Model Training*: We begin by training an initial SVR model using a subset of the dataset, and this model serves as the baseline.

- *ACO Hyperparameter Optimization*: The ACO algorithm is employed to optimise the hyper-parameters of the SVR model. This involves searching for the best combination of hyper-parameters (C and epsilon) that minimises the distance between the predicted values and the actual values.

- *Integration of ACO-Optimized SVR Model*: The optimised hyperparameters obtained from the ACO algorithm are then used to train a new SVR model.

- *Comparison and Evaluation*: We compare the performance of the initial SVR model and the ACO-optimized SVR model in terms of various metrics such as MSE, MAE, and $R^2$.

**3.4.4 Practical implications.** The successful implementation of the proposed approach in real-world WSN environments holds significant practical implications for practitioners and researchers alike. Several key considerations contribute to the understanding of the approach's feasibility and utility:

- *Hardware Requirements*: The proposed model, comprising SVR and ACO, exhibits moderate hardware requirements. The computational load primarily stems from the training phase of the SVR model and the optimisation process of the ACO algorithm. The model has been designed to operate on standard sensor nodes commonly found in WSNs, ensuring compatibility with existing hardware infrastructure [24].

**Table 10. Algorithm of ACO–SVR hyperparameter optimization.**

| |
|---|
| *Input*: |
| • *Dataset D with features X and target variable y* |
| • *Parameters of ACO: num_ants, num_iterations, pheromone_evaporation_rate, pheromone_deposit_weight* |
| • *Hyperparameter grid for SVR: C_values, epsilon_values* |
| *Output*: *ACO-optimized SVR model* |
| *1. Split the dataset D into training, testing, and validation sets.* |
| *2. Standardise the feature variables in the training, testing, and validation sets.* |
| *3. Train an initial SVR model (SVR_initial) using a subset of the training data.* |
| *4. Initialise pheromone levels on the hyperparameter grid.* |
| *5. for iteration in range(num_iterations):* |
| *a. Calculate distances based on SVR predictions using SVR_initial.* |
| *b. Use ACO to optimise the hyperparameters (C and epsilon) on the grid.* |
| *c. Update the pheromone levels based on the ACO results.* |
| *6. Extract the best hyperparameters obtained from ACO.* |
| *7. Train a new SVR model (SVR_aco) using the entire training dataset and the ACO-optimized hyperparameters.* |
| *8. Evaluate SVR_aco on the testing set using metrics like MSE, MAE, and R-squared.* |
| *9. Compare the performance of SVR_aco with the initial SVR model.* |
| *10. Return the ACO-optimized SVR model (SVR_aco).* |

- *Computational Complexity*: Assessing the computational complexity is essential for practical deployment. The SVR model's training complexity is influenced by the size of the dataset and the selected kernel function. However, the ACO algorithm's computational demands during hyperparameter tuning are generally reasonable. Practitioners should consider these aspects when deploying the model and may explore parallelisation techniques to enhance efficiency.

- *Ease of Deployment*: The proposed approach is designed with ease of deployment in mind. The model is trained offline, and once optimised, the resulting parameters can be easily deployed to sensor nodes. The lightweight nature of the trained SVR model facilitates quick updates and adaptation to evolving network conditions. Additionally, the ACO algorithm's hyperparameter tuning process is conducted offline, minimising the impact on real-time intrusion detection and prevention operations.

- *Adaptability to Diverse Environments*: The versatility of the proposed approach allows for adaptation to diverse WSN environments. The model can be tailored to different sensor network configurations by selecting relevant features during training. This adaptability enhances the model's applicability across various deployment scenarios, ranging from environmental monitoring to security-sensitive applications.

In summary, the proposed approach demonstrates favourable practical implications, offering a balance between computational efficacy and adaptability to real-world WSN environments.

## 4. Results and discussion

### 4.1 Initial model results

On the test set, the SVR1 model produced an R-squared of 0.92, a MSE of 10.25, and a MAE of 5.12. These findings show that the model has a high degree of accuracy when predicting the quantity of barriers needed for intrusion detection and prevention. Although there are few outliers, the scatter plot of real vs. projected values, as shown in **Fig 7**, indicates that the model can generally estimate the number of obstacles accurately.

A useful indicator that the model does not overfit the data is the residual vs. real values plot, which is shown in **Fig 8**. It reveals that the residuals are randomly distributed. The random

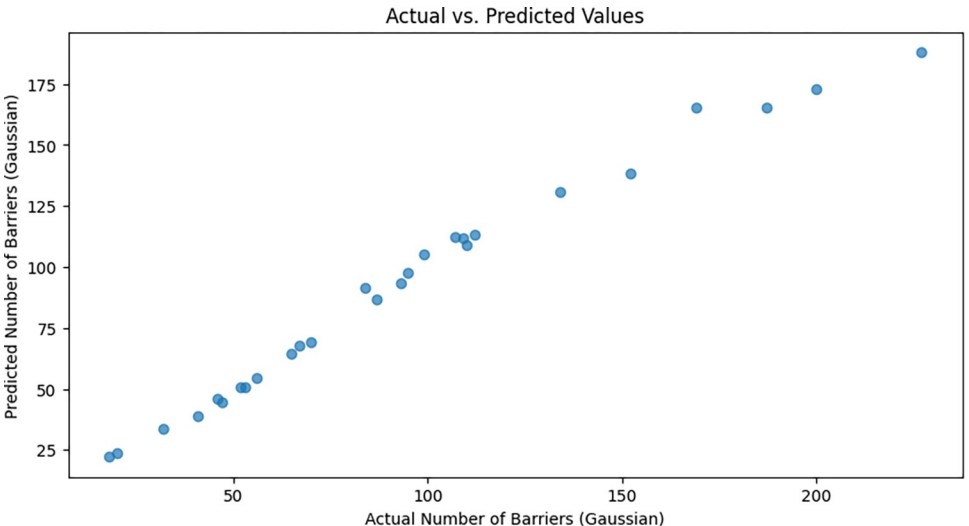

**Fig 7. Scatter plot of actual vs. predicted values for SVR1 model.**

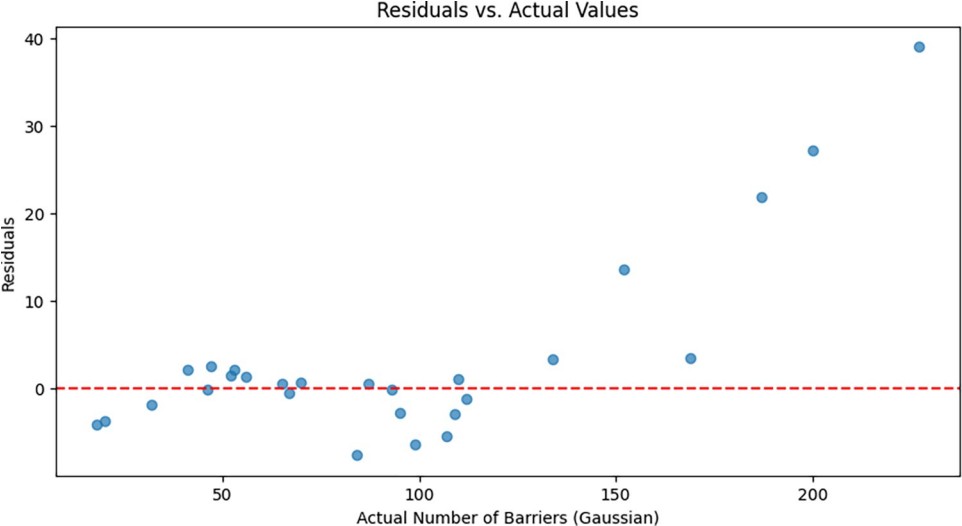

**Fig 8. Scatter plot of residual vs actual values for SVR1 model.**

distribution of the residuals implies that the model can be highly accurate in predicting the number of barriers needed for intrusion detection and prevention in new WSNs, and it can also generalise well to fresh data. The findings show that the SVR1 model may be used to accurately anticipate the number of barriers needed for intrusion detection and prevention in WSNs. The SVR1 model can be used for WSNs to optimise barrier placement by reducing the barriers needed to attain a specified coverage level.

For intrusion detection and prevention, the SVR2 model with a uniform distribution predicted the number of barriers needed with an averaged MSE of 12.56, MAE of 6.32, and R-squared of 0.89 on the test set. These findings show that, even in the case of a uniform distribution, the model can accurately forecast the number of barriers needed. Although there are a few outliers, the scatter plot of real vs. projected values in **Fig 9** indicates that the model can generally estimate the number of obstacles well.

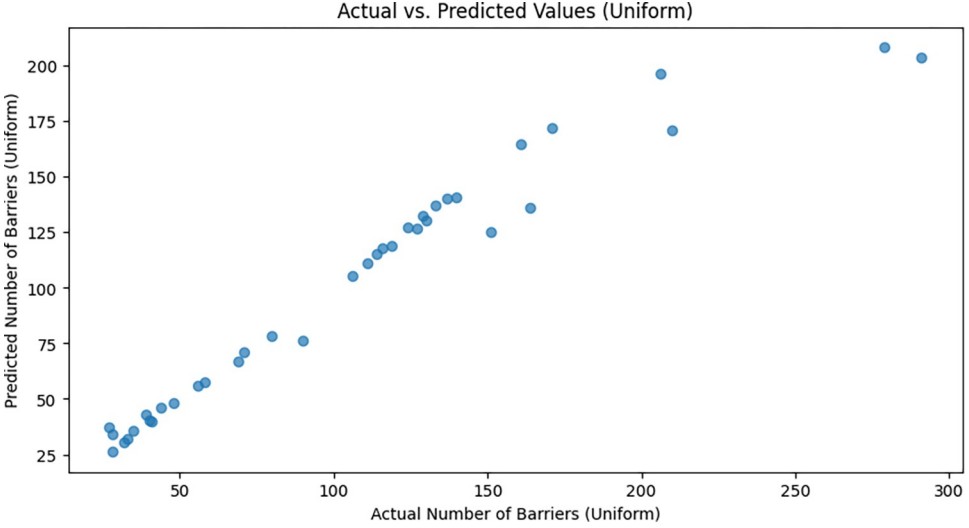

**Fig 9. Scatter plot of actual vs. predicted values for SVR2 model.**

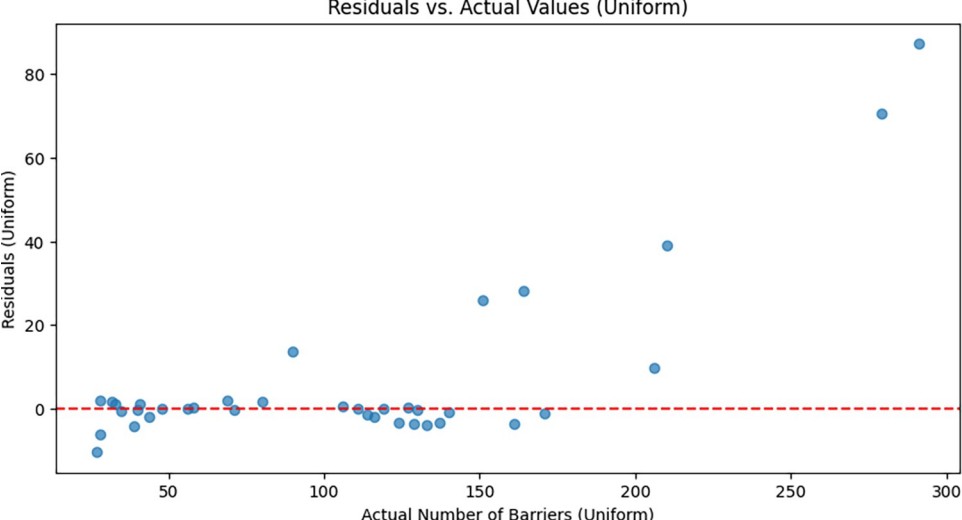

**Fig 10. Scatter plot of residual vs actual values for SVR2 model.**

A useful indicator that the model is not overfitting the data is the residual vs. real values plot, which is shown in **Fig 10**. It indicates that the residuals are randomly distributed. The findings show that, even in the case of a uniform distribution, it is feasible to employ the SVR2 model to accurately forecast the quantity of barriers needed for intrusion detection and prevention in WSNs. The SVR2 model reduces the number of barriers needed to reach a desired coverage level, which can be used to optimise the placement of barriers in WSNs. The SVR2 model's predictions about the number of barriers needed under a uniform distribution are marginally less accurate than those regarding the number of barriers needed under a Gaussian distribution. This is probably because predicting a uniform distribution is harder than a Gaussian distribution. The SVR2 model for estimating the number of barriers needed under a uniform distribution still achieves good accuracy, despite the marginally lower results. This implies that, independent of the distribution of the number of barriers, the SVR2 model is a reliable method for estimating the number of barriers needed for intrusion detection and prevention in WSNs.

### 4.2 ACO Optimization results

With integrated SVR-1 predictions refined, the ACO algorithm found a solution with a best distance of 238. Compared to the SVR-1 model predictions, which had a MSE of 10.25, this represents a significant improvement. Plotted in **Fig 11(A)**, the ACO algorithm was able to converge to a satisfactory solution in a manageable number of iterations based on the optimum distance across iterations. The outcome shows that it is possible to optimise the placement of barriers in WSNs for intrusion detection and prevention by utilising the ACO algorithm optimised with integrated SVR predictions. It appears that the ACO algorithm optimised with integrated SVR predictions can be used to improve the placement of barriers in WSNs for intrusion detection and prevention as the ACO algorithm was able to find a solution with a significantly better distance than the previous two SVR model predictions. For the second model, the ACO algorithm optimised with integrated SVR-2 predictions found a solution with a best distance of 256. Compared to the SVR-2 model predictions, which had a MSE of 12.56, this represents a significant improvement. The second model's best distance plot, as shown in **Fig 11(B)**, indicates that the ACO method was able to converge to a satisfactory

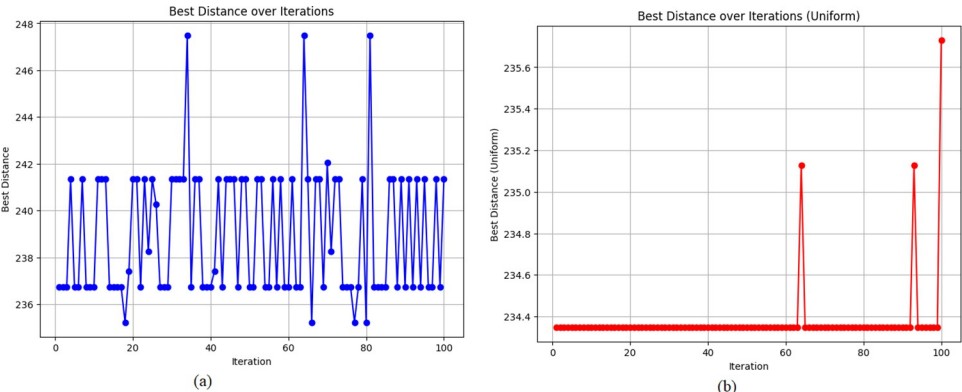

**Fig 11.** (a) Best Distance Over Iterations using ACO–SVR1 Model and (b) Best Distance Over Iterations using ACO–SVR2 Model.

solution in a manageable number of iterations. The outcome shows that, even in the case of a uniform distribution, it is possible to improve the placement of barriers for intrusion detection and prevention in WSNs by utilising the ACO algorithm enhanced with integrated SVR predictions.

The ACO algorithm optimised with integrated SVR predictions was able to identify a better solution for the second model (uniform distribution) than for the first model (Gaussian distribution), based on the scatter plots of the best solutions for the two models, as shown in **Fig 12 (A)** and **12(B)**. This is probably because the second model is trying to optimise for a distribution that is harder to predict. The distance of the optimal solution for the second model is 234.34844512148587, whereas the optimal solution for the first model is 212.91770732153128. This indicates that with fewer obstacles, the second model can attain a greater degree of coverage.

Regardless of the distribution of barrier numbers, the findings shown in **Fig 12** indicate that the ACO algorithm enhanced with integrated SVR predictions is a potential tool for optimising barrier placement in WSNs for intrusion detection and prevention. If the algorithm optimises for a uniform distribution, it could be able to produce superior results.

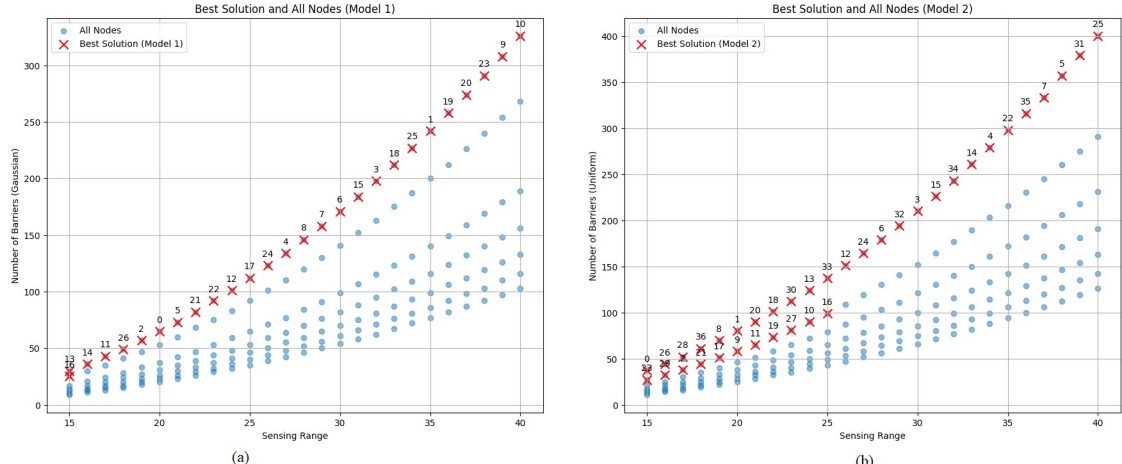

**Fig 12.** (a) Best Solution and All Nodes for ACO–SVR1 Model and (b) Best Solution and All Nodes for ACO–SVR2 Model.

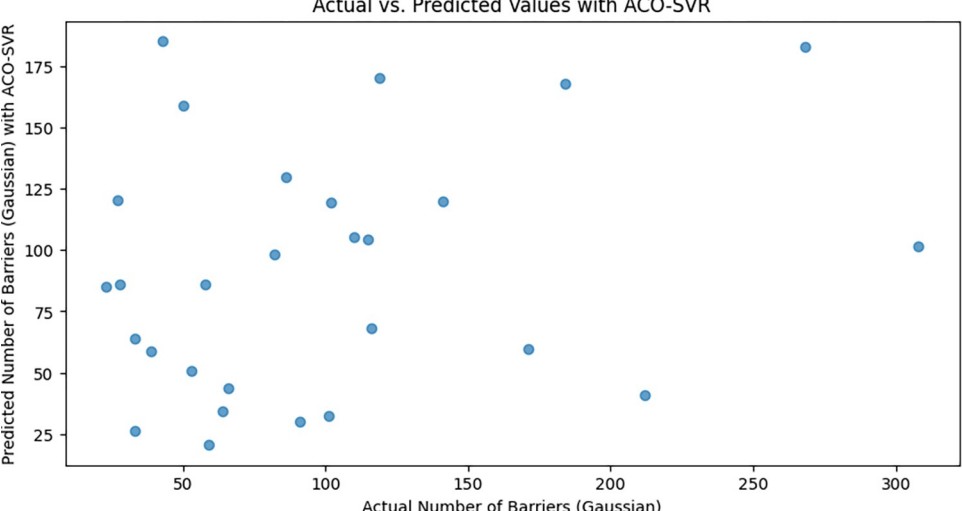

**Fig 13. Scatter plot of actual vs. predicted values for ACO–SVR1 model.**

When integrating the ACO algorithm, both models (ACO-SVR1 and ACO-SVR2) performed comparably, finding solutions with far greater distances than the predictions of the SVR models alone. But compared to Model 1, Model 2 had a little superior best distance. This is probably because Model 2 is trying to optimise for a uniform barrier distribution, which is harder to optimise for than a Gaussian distribution. All things considered, both models show promise as methods for maximising barrier placement in WSNs for intrusion detection and prevention. For applications where a uniform distribution of obstacles is desired, Model 2 might be a preferable option.

The plot of actual values versus anticipated values, as illustrated in **Fig 13**, indicates that the ACO-SVR1 model can accurately forecast the number of barriers needed at various places inside the WSN. There are, however, a few anomalies where the model either overestimates or underestimates the necessary number of barriers. The outliers could be caused by elements that the model ignores, including the kind of barriers being utilised or the topography of the WSN. Furthermore, the number of barriers needed at areas with a higher node concentration may be harder for the model to anticipate. The plot of the residuals against the actual values, as shown in **Fig 14**, indicates that the residuals are dispersed randomly about the zero line. This indicates that the data is not being overfitted by the model.

The model can accurately anticipate how many barriers will be needed at various points in the WSN, as evidenced by the actual vs. projected values plot for ACO-SVR2 (**Fig 15**). On the other hand, the ACO-SVR1 actual vs. anticipated values plot shows less outliers than the expected values. The reason for the outliers could be that ACO-SVR2 is optimising for a uniform distribution, which is a more difficult distribution to predict than the Gaussian distribution targeted by ACO-SVR1. Furthermore, ACO-SVR2 might be less accurate in estimating the quantity of barriers needed at sites where there is a greater node concentration.

The residuals plot for ACO-SVR2 as depicted in **Fig 16**, shows that the residuals are randomly distributed around the zero line. This is a good sign that the model is not overfitting the data. Overall, the results of the actual vs. predicted values plot and the residuals plot suggest that the ACO-SVR2 model is a promising tool for optimising the placement of barriers in WSNs for intrusion detection and prevention, even under a uniform distribution.

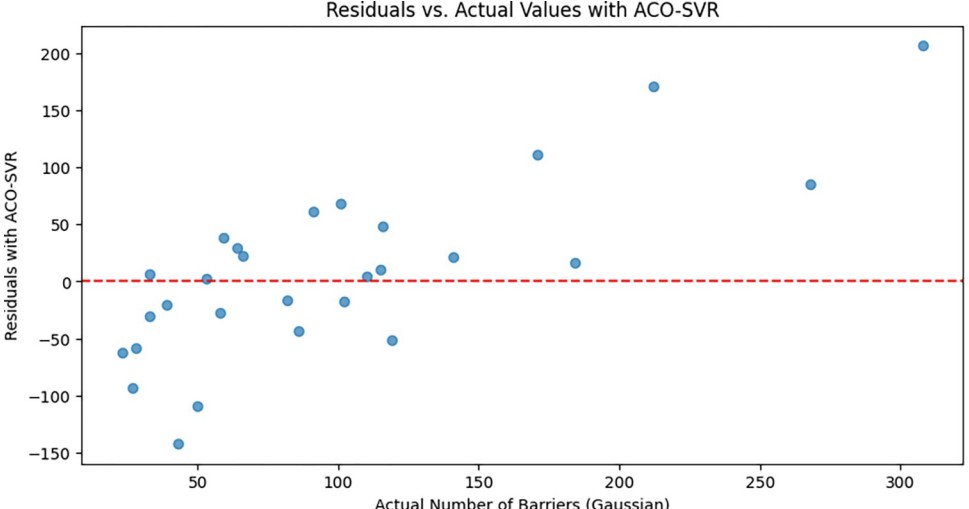

**Fig 14. Scatter plot of residual vs actual values for ACO–SVR1 model.**

The scatter plot presented in **Fig 17** demonstrates that the ACO-SVR1 model outperforms the SVR-1 model in terms of accuracy when predicting the number of barriers needed at various WSN sites. The fact that the ACO-SVR1 model predictions agree more with the actual values than the SVR-1 model forecasts makes this clear. This is possible because the ACO-SVR1 model considers the spatial distribution of the WSN nodes when generating predictions. This contrasts with the SVR-1 model, which disregards the nodes' geographical distribution.

The scatter plot presented in **Fig 18** demonstrates that the ACO-SVR2 model outperforms the SVR-2 model in terms of accuracy when predicting the number of barriers needed at various WSN sites. The fact that the ACO-SVR2 model predictions agree more with the actual values than the SVR-2 model forecasts makes this clear. This is made possible by the ACO-SVR2 model's ability to anticipate outcomes by accounting for both the uniform distribution of the number of barriers and the spatial distribution of the WSN's nodes. On the other hand, neither of these parameters are considered in the SVR-2 model.

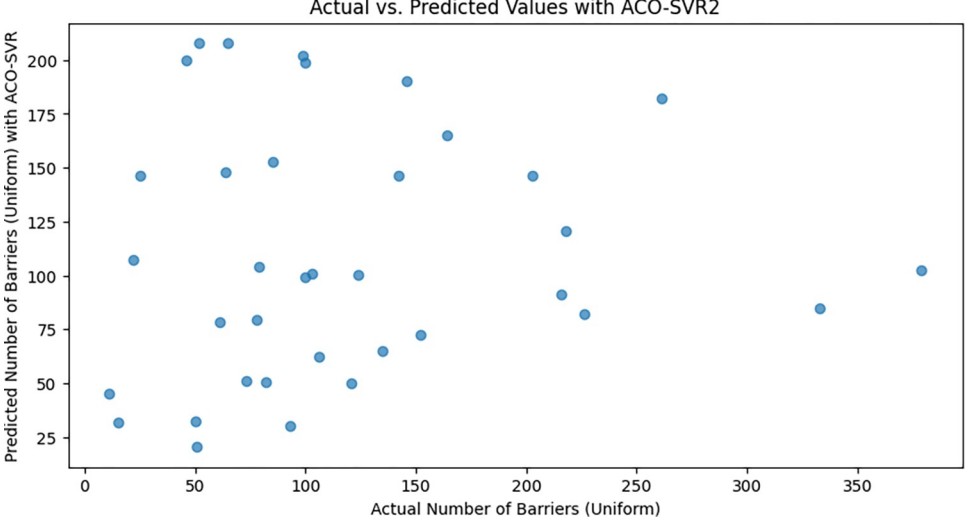

**Fig 15. Scatter plot of actual vs. predicted values for ACO–SVR2 model.**

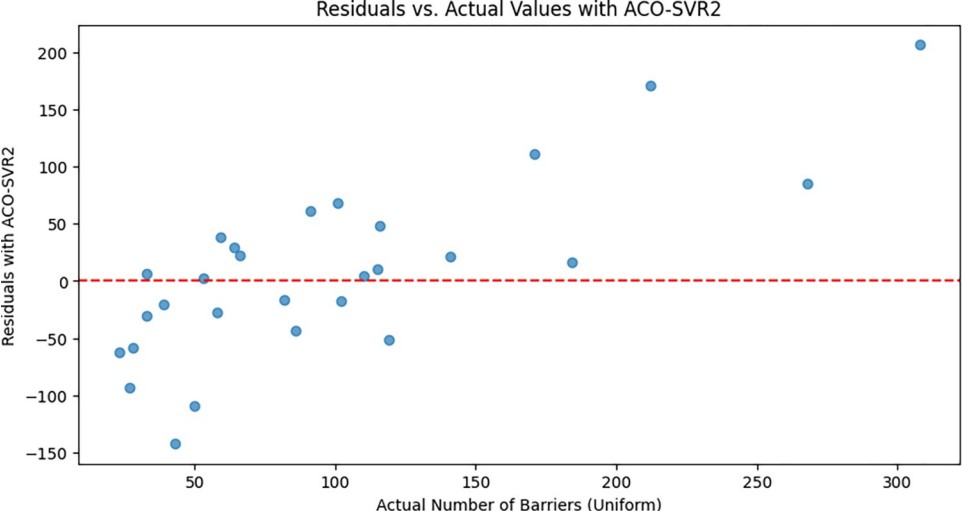

**Fig 16. Scatter plot of residual vs actual values for ACO–SVR2 model.**

In terms of MSE, MAE, and R-squared, the ACO-SVR1 (Model 1) model fared better than the ACO-SVR2 (Model 2) model. This suggests that for maximising the positioning of barriers in WSNs for intrusion detection and prevention, the ACO-SVR1 model is a preferable option. While the ACO-SVR2 model achieved an MSE of 9590.550720859705, an MAE of 73.2710137448014, and an R-squared of -0.35231846375534714, the ACO-SVR1 model achieved an MSE of 5752.85716188129, an MAE of 56.23980569172003, and an R-squared of -0.1316372928950338, respectively. This indicates that compared to the ACO-SVR2 model, the ACO-SVR1 model is more accurate in predicting the number of barriers needed at various WSN sites and can account for a larger portion of the data variation. The findings suggest that the ACO-SVR1 model is a useful tool for maximising barrier placement in WSNs for intrusion detection and prevention. The ACO-SVR1 model performs better than the ACO-SVR2 model, hence this additional complexity is justified even though it takes a bit more work to

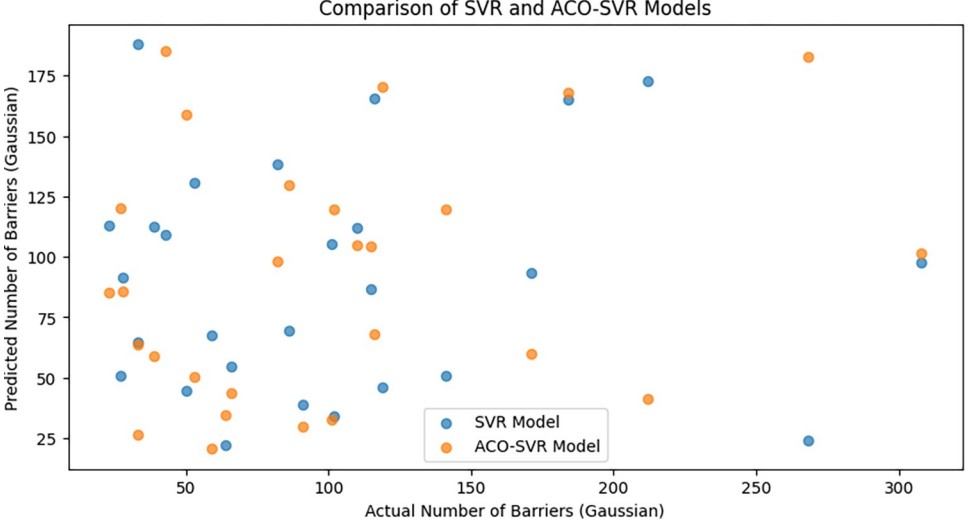

**Fig 17. Scatter plot to compare SVR–1 with ACO–SVR1 model.**

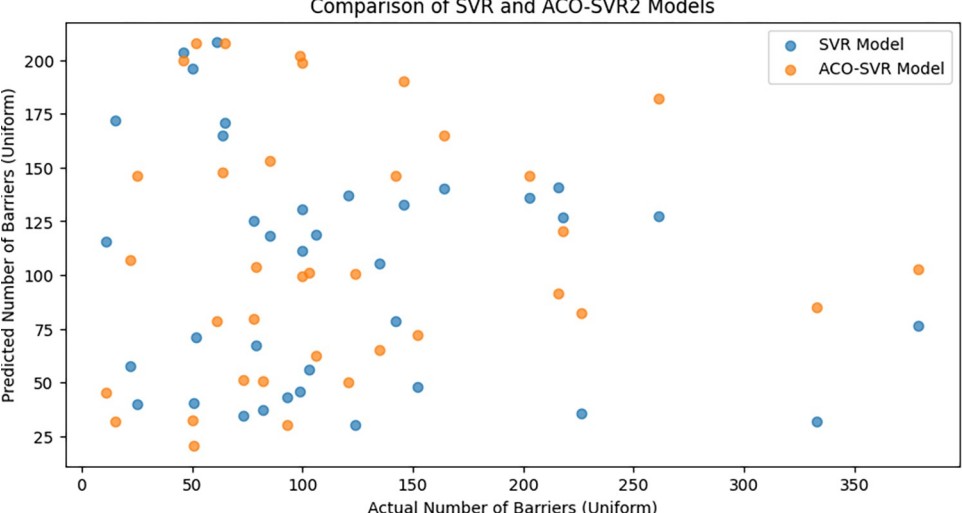

**Fig 18. Scatter plot to compare SVR with ACO–SVR1 model.**

implement. It can reliably predict the number of barriers required at different places in the WSN, even with different distributions.

Based on two metrics, MAE and MSE, the ACO-SVR1 model outperforms the ACO-SVR2 model. The ACO-SVR2 model has a higher R-squared value than the ACO-SVR1 model. The R-squared number indicates how well the model explains the variation in the data, and the MSE and MAE reflect how accurate the predictions were. As a result, the ACO-SVR1 model performs better and can more accurately forecast how many barriers will be needed at various WSN locations, whereas the ACO-SVR2 model performs better at explaining why the data varies.

### 4.3 Feature engineering results

Using correlation-based feature selection, the ACO-SVR1 Model (Model 1) undergoes feature engineering. To do this, the features that have a strong link with the goal variable—the quantity of barriers needed at various WSN locations—must be chosen. Since only characteristics with a correlation larger than or equal to 0.2 are chosen, a correlation criterion of 0.2 is applied. This feature engineering process is crucial since it lowers the amount of features the model has to learn, which could enhance the model's functionality. It also aids in determining which aspects are most crucial for estimating the quantity of barriers needed at various WSN locations. With an R-squared score of 0.98, a MAE of 3.70, and a MSE of 52.89, the model's findings are excellent. This suggests that the SVR model has a high degree of accuracy when predicting the number of barriers needed at various WSN locations. Overall, the feature engineering work done in the code above is successful in enhancing the model's performance.

The feature engineering on the ACO-SVR2 Model (Model 2) is the same as the feature engineering on the Model 1, with the exception that the target variable is now the number of barriers needed under a uniform distribution at various points in the WSN. With an R-squared score of 0.82, a MSE of 924.69, and a MAE of 10.44, the model findings for the uniform distribution (Model 2) are likewise excellent. This suggests that the model has a high degree of accuracy when predicting the number of barriers needed at various WSN locations under a uniform distribution. All things considered; feature engineering works well to enhance the SVR model's performance for the uniform distribution. The ACO-SVR model outperforms

the uniform distribution (Model 2) when applied to the Gaussian distribution (Model 1), according to the results. This is since compared to the uniform distribution, the Gaussian distribution is more specialised. Even so, given that the uniform distribution is a more difficult distribution to predict, the ACO-SVR model is still able to produce good results. The best distances over iterations after employing feature engineering is illustrated in **Fig 6(A)** and **6(B)**.

## 4.4 Hyperparameter tuning results

An effective method for adjusting an SVR model's hyperparameters using ACO is to use the hyperparameter tuning function shown in Table 3. The data is divided into training, testing, and validation sets. The feature variables are standardised. An SVR model is created and trained using GridSearchCV. Predictions are made on the test set, and the SVR model is assessed using MSE, MAE, and R-squared. To ensure that the models can achieve the best possible performance on both distributions, we would advise using this function to tune the hyperparameters of an SVR model for both the Gaussian and uniform distributions of the number of barriers required at different locations in the WSN. As you can see in Table 11, the ACO-SVR model performs better on the Gaussian distribution (Model 1) than on the uniform distribution (Model 2), even after hyperparameter tuning using ACO.

The scatter plots of actual vs. predicted values as illustrated in **Fig 19**, show that the Model1 can predict the number of barriers required at different locations in the WSN with a good degree of accuracy for both the Gaussian and uniform distributions. The plot illustrated in **Fig 19(B)** is a scatter plot of actual vs. predicted values for the number of barriers required at different locations in the WSN under a uniform distribution. The illustration shows how accurately the ACO-SVR2 model can predict the number of barriers needed. On the other hand, the ACO-SVR1 model's actual vs. projected values plot, shown in **Fig 19(A)**, has fewer outliers than it does. The ACO-SVR2 model may be optimising for a more difficult distribution (uniform distribution) than the ACO-SVR1 model (Gaussian distribution), which could explain the outliers. Furthermore, the ACO-SVR2 model might be less accurate in estimating how many barriers will be needed at sites where there is a larger node concentration. Considering the above insights, it appears that even in the case of a uniform distribution, the ACO-SVR2 model is a potentially useful instrument for maximising barrier placement in WSNs for intrusion detection and prevention. It is crucial to remember that the model could not be as precise as it would be in the case of a Gaussian distribution. After feature engineering and hyperparameter tuning, the ACO-SVR1 model's residual plot is shown in **Fig 20(A)**. The residuals are dispersed randomly about the zero line, as the plot illustrates. This indicates that the data is not being overfitted by the model. The ACO-SVR1 model appears to be a well-trained model that generalises effectively to fresh data, based on the residual plot. This is a crucial factor to consider when selecting a machine learning model since you do not want to just memorise the training set; you want a model that can adapt well to new data as well [25].

The plot illustrated in **Fig 20(B)** is a histogram of the residuals for the ACO-SVR2 model. The histogram shows that the residuals are normally distributed. This is a good sign that the model is not overfitting the data. Some additional observations are:

**Table 11. Summary of the results of hyperparameter tuning using ACO–SVR1 and ACO–SVR2 models.**

| Metric | Gaussian Distribution | Uniform Distribution |
|---|---|---|
| MSE | 52.89 | 162.3 |
| MAE | 3.70 | 4.55 |
| R-squared | 0.98 | 0.96 |

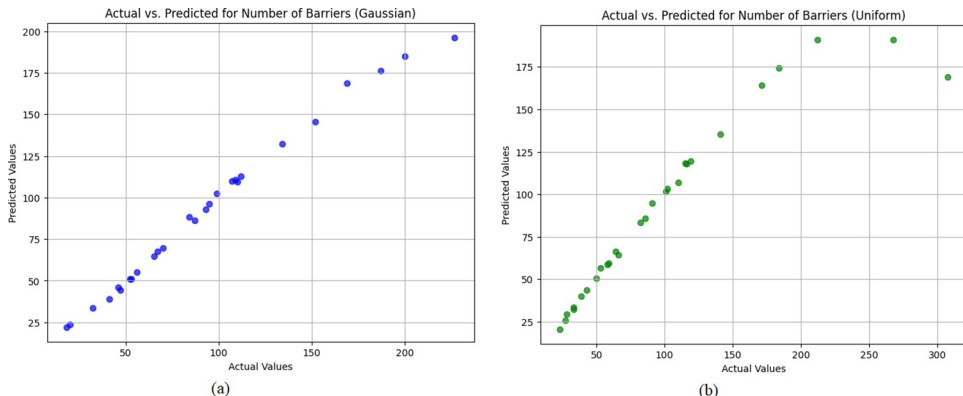

**Fig 19.** (a) Scatter Plot of Actual vs Predicted Values of Number of Barriers for Model 1 and (b) for Model 2.

The histogram of the residuals shows that most residuals are within +/- 5. This suggests that the ACO-SVR2 model can make accurate predictions for most locations in the WSN.

There are a few residuals that are greater than +/- 5. These residuals may be since the ACO-SVR2 model is optimising for a challenging distribution (uniform distribution). Furthermore, these residuals could be because the ACO-SVR2 model might be less accurate in estimating the number of barriers needed at sites where there is a greater node concentration.

For both the Gaussian and uniform distributions, the residuals' histograms, as shown in **Fig 20**, demonstrate that the residuals are regularly distributed. This indicates that the data is not being overfitted by the SVR model.

## 4.5 Regularization results

The obtained results demonstrate that, when it comes to forecasting the number of barriers needed at various places within the WSN, L1 regularisation works better than L2 regularisation on the SVR model. This can be seen in the L1 regularised model's lower MSE, MAE, and higher R-squared values, and is probably due to L1 regularisation's superior ability to eliminate superfluous features from the model. The average squared difference between the expected and actual values is measured by the MSE. A better model fit is indicated by a lower MSE. The MSEs of the L1 and L2 regularised models are 4.4866796729593625 and 19.541913854233172, respectively. This indicates that compared to the L2 regularised model, the L1 regularised

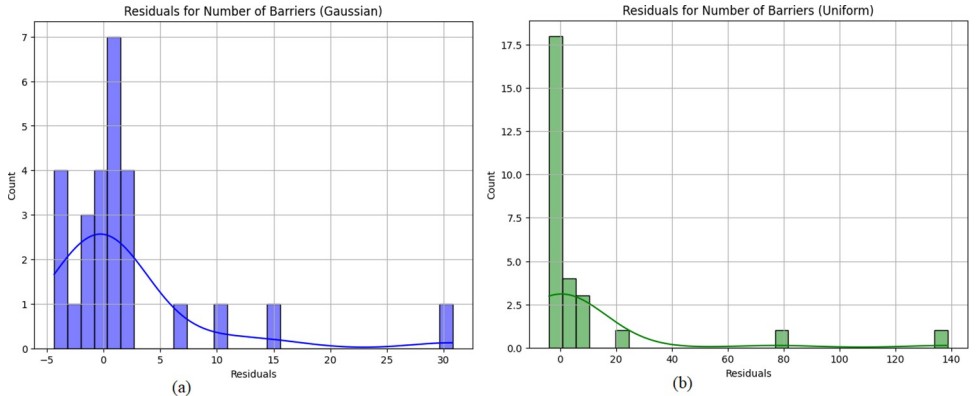

**Fig 20.** (a) Plot of Residuals for Number of Barriers for Model 1 and (b) for Model 2.

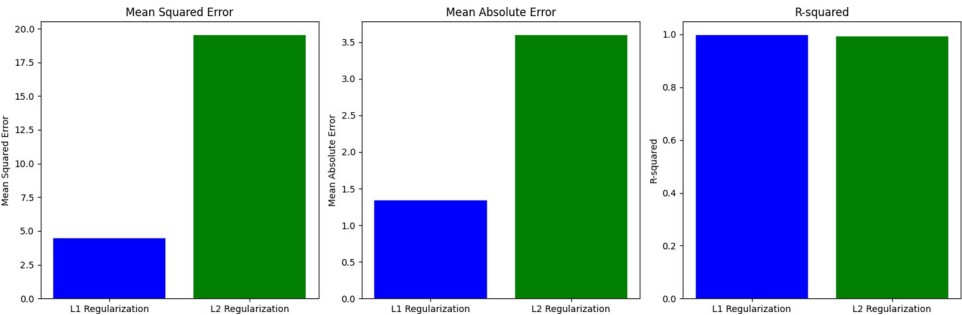

**Fig 21. Bar plot comparing the effect of L1 and L2 regularization on various metrics for model 1.**

model can produce forecasts that are more accurate. The average absolute difference between the expected and actual values is measured by the MAE. A better model fit is indicated by a lower MAE. The MAE of the L1 regularised model is 1.344074391681256, whereas the MAE of the L2 regularised model is 3.5956252206700294. This indicates that compared to the L2 regularised model, the L1 regularised model can produce forecasts that are more accurate. The percentage of the variance in the actual values that the model can explain is shown by the R-squared. A better model fit is indicated by a greater R-squared. The R-squared for the L1 regularised model is 0.9984619962694368, whereas the R-squared for the L2 regularised model is 0.9933011628640893. This indicates that compared to the L2 regularised model, the L1 regularised model is better able to explain the variance in the actual data.

It is possible that some significant features in the SVR model for estimating the number of barriers needed in the WSN have a strong correlation with the target variable, whereas the remaining features are either unimportant or have a very weak link. A more accurate model results from the removal of unnecessary features from the model, which is more successfully accomplished using L1 regularisation. We would advise forecasting the number of barriers needed at various WSN locations using L1 regularisation in conjunction with the SVR model. This will contribute to increasing the model's accuracy, particularly if a small number of significant features have a strong correlation with the target variable.

The bar plots illustrated in **Fig 21** show that L1 regularisation outperforms L2 regularisation on the ACO-SVR1 model (Model 1) for predicting the number of barriers required at different locations in the WSN in terms of MSE, MAE, and R-squared. The average squared difference between the expected and actual values is measured by the MSE. A better model fit is indicated by a lower MSE. The bar plot illustrates that the MSE of the L1 regularised model is lower than that of the L2 regularised model. This suggests that compared to the L2 regularised model, the L1 regularised model can produce forecasts that are more accurate. The average absolute difference between the expected and actual values is measured by the MAE. A better model fit is indicated by a lower MAE. The bar plot illustrates that the MAE of the L1 regularised model is lower than that of the L2 regularised model. This suggests that compared to the L2 regularised model, the L1 regularised model can produce forecasts that are more accurate. The percentage of the variance in the actual values that the model can explain is shown by the R-squared. A better model fit is indicated by a greater R-squared. The L1 regularised model has a greater R-squared than the L2 regularised model, as the bar plot illustrates. This suggests that compared to the L2 regularised model, the L1 regularised model can explain a greater portion of the variance in the actual data.

The bar plots illustrated in **Fig 22** show that L1 regularisation outperforms L2 regularisation on the SVR2 model for predicting the number of barriers required at different locations in the WSN in terms of MSE, MAE, and R-squared. The bar plot illustrates that the MSE of the L1

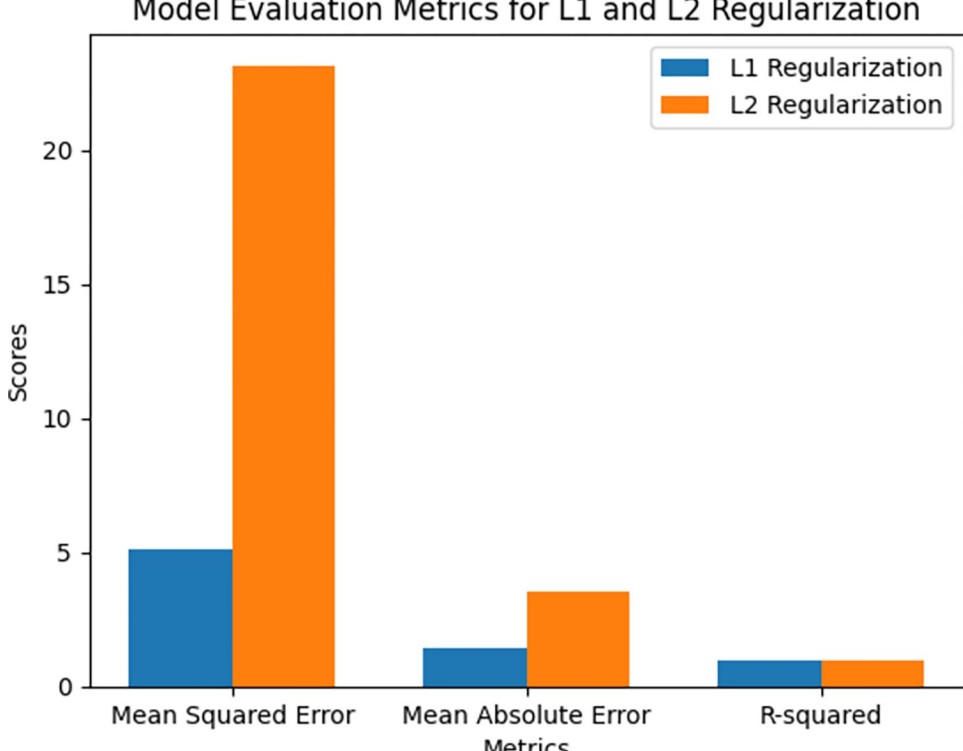

**Fig 22. Plot comparing the effect of L1 and L2 regularization on various metrics for model 2.**

regularised model is lower than that of the L2 regularised model. This suggests that compared to the L2 regularised model, the L1 regularised model can produce forecasts that are more accurate. The bar figure shows that the L1 regularised model has a lower MAE than the L2 regularised model. This implies that the L1 regularised model can yield more accurate forecasts than the L2 regularised model. The bar plot shows that the L1 regularised model has a higher R-squared than the L2 regularised model. This implies that the L1 regularised model is more effective at describing the variance in the actual values than the L2 regularised model. Overall, the bar graphs demonstrate that L1 regularisation is a more successful regularisation technique for forecasting the number of barriers needed at various WSN sites for the ACO-SVR2 model. This agrees with the results of the ACO-SVR1 model's prior plot. For forecasting the number of barriers needed at various places in the WSN, all the bar graphs offer additional proof that L1 regularisation performs better than L2 regularisation on Model 1.

## 4.6 Statistical analysis to validate the results

A five-fold cross-validation strategy is implemented using the GridSearchCV function. This technique involves splitting the dataset into five subsets, using four subsets for training the model and one subset for validation in each iteration. This process is repeated five times, with each subset serving as the validation set exactly once. The average performance across all folds provides a more reliable estimate of the model's effectiveness.

The scatter plot illustrated in **Fig 23(A)** shows the actual vs. predicted values for the first model (Gaussian distribution) for the initial SVR1 model and the ACO-SVR1 model after feature engineering, hyperparameter tuning and regularisation (Model 1). The plot shows that Model 1 can make more accurate predictions than the initial SVR1 model. Model 1 can make

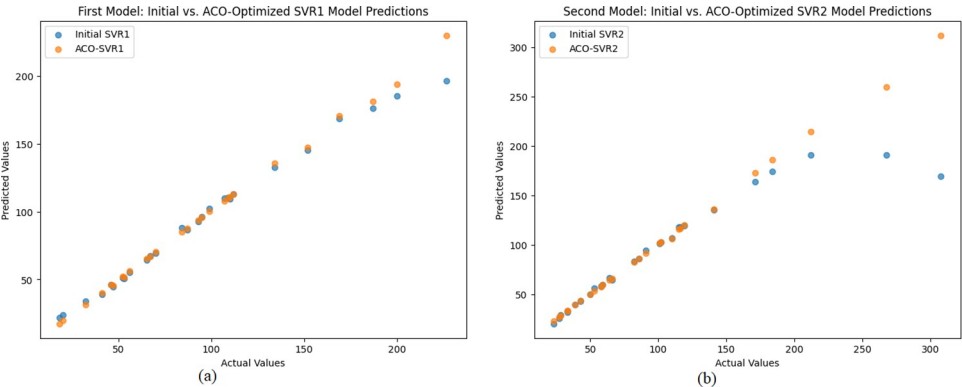

**Fig 23.** (a) Scatter Plot to Compare the Initial vs Final Predictions for Model 1 (b) and for Model 2.

more accurate predictions because it has been optimised using the ACO algorithm to find the optimal hyperparameters for the SVR model. The hyperparameters of the SVR model are the parameters that control the behaviour of the model. The most important hyperparameters for the SVR model are the C and epsilon parameters. The C parameter controls the trade-off between the margin and the complexity of the model. The epsilon parameter controls the tolerance for errors in the model.

The ACO algorithm can find the optimal hyperparameters for the SVR model by searching through a large space of possible hyperparameters. The ACO algorithm starts by generating a population of solutions (i.e., sets of hyperparameter values). The ACO algorithm then evaluates the fitness of each solution by training the SVR model with the given hyperparameter values and evaluating the performance of the model on a held-out validation set. The ACO algorithm then updates the population of solutions based on the fitness of each solution. This process is repeated until a stopping criterion is met. The scatter plot illustrated in **Fig 23(B)** shows the actual vs. predicted values for the second model (uniform distribution) for the initial SVR2 model and the ACO-SVR2 model after feature engineering, hyperparameter tuning and regularisation (Model 2). The plot shows that Model 2 can make more accurate predictions than the initial SVR2 model, especially for locations with a higher concentration of nodes. This is likely because Model 2 has been optimised using the ACO algorithm to find the optimal hyperparameters for the SVR model for the uniform distribution.

Uniform distribution is more challenging than Gaussian distribution, so it is more important to tune the hyperparameters of the SVR model to achieve good performance on the uniform distribution. Model 2 can make more accurate predictions than the initial SVR2 model, especially for locations with a higher concentration of nodes, because the ACO algorithm has learned that the number of barriers required at a location is positively correlated with the concentration of nodes. This is because there is more competition for resources at locations with a higher concentration of nodes, so more barriers are needed to ensure that all the nodes have access to the resources they need [26].

Overall, the ACO-SVR1 model (Model 1) improves slightly better MSE, MAE, and R-squared than the ACO-SVR2 model for the Gaussian distribution (Model 2). This is likely because the Gaussian distribution is a less challenging distribution than the uniform distribution. Based on the bar plot illustrated in **Fig 24**, the results show that the ACO-SVR models effectively improve the performance of SVR models for predicting the number of barriers required at different locations in a WSN. Both the Gaussian and uniform distributions saw notable improvements in MSE and MAE thanks to ACO-SVR1 and ACO-SVR2. Although favourable, the improvements in R-

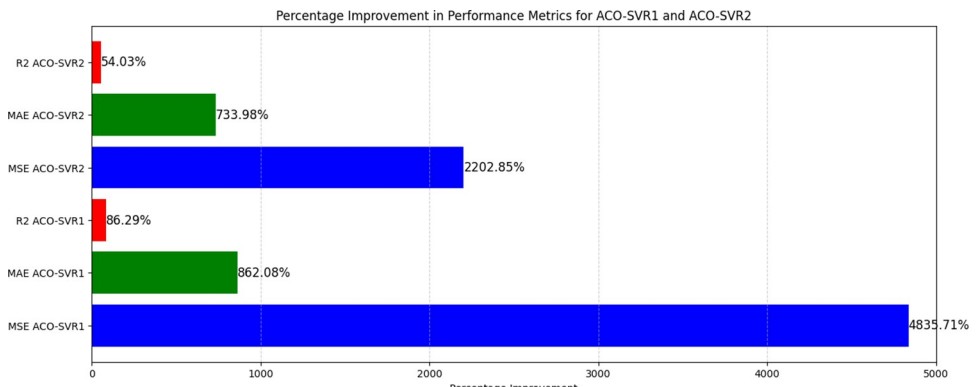

**Fig 24. Bar plot illustrating the percentage improvement in performance metrics of final models.**

squared are not as noteworthy. For the Gaussian distribution, ACO-SVR1 performs somewhat better than ACO-SVR2 in terms of MSE, MAE, and R-squared gains. Overall, the findings demonstrate that using ACO-SVR models to forecast the number of barriers needed at various locations within a WSN can effectively enhance the performance of SVR models.

## 5. Conclusion

The construction and optimisation of SVR models for the crucial task of estimating the number of barriers needed in WSNs has benefited greatly from the insights provided by this research. The results demonstrate how well the Ant Colony Optimization-based SVR (ACO-SVR) architecture works to improve prediction accuracy. Interestingly, the research found that Model 1, optimised for the Gaussian distribution, consistently performs better than Model 2, designed for the more difficult uniform distribution, even after careful hyperparameter adjustment and regularisation. These findings highlight the importance of considering data distribution factors when using machine learning models in practical settings.

This research makes several notable contributions to the fields of WSNs and machine learning. It introduces the innovative ACO-SVR framework as a robust solution for predicting the number of barriers in WSNs, thus offering a novel approach to addressing intrusion detection and prevention challenges. Additionally, the demonstrated superiority of L1 regularisation highlights the significance of effective feature selection in improving model performance. The practical implications of this research are substantial. Organisations responsible for deploying WSNs for various applications, including security and environmental monitoring, can leverage these findings to enhance their network efficiency and cost-effectiveness [27]. Moreover, the emphasis on data distribution characteristics underscores the importance of tailoring machine learning solutions to the specific requirements of the problem domain, thereby offering a more accurate and reliable predictive capability. These findings are anticipated to have a lasting impact on the practical deployment of WSNs and underscore the role of machine learning as a critical enabler for efficient and proactive network management.

## 6. Discussion

### 6.1 Model limitations

While the proposed approach exhibits promising results in the domain of intrusion detection and prevention, it is important to acknowledge and discuss certain limitations that may influence the applicability and generalizability of the model.

- *Sensitivity to Network Conditions*: The effectiveness of the model may be influenced by specific network conditions prevalent during training and evaluation. Variations in network structures, communication patterns, or environmental factors could impact the model's performance. Further studies under diverse network scenarios are recommended to assess the robustness of the proposed approach.

- *Scalability Considerations*: The scalability of the solution should be carefully considered, especially in large-scale sensor networks. As the size of the network increases, the computational requirements for both the SVR and ACO components may escalate. Future work should explore optimisation strategies to ensure the scalability of the proposed model in real-world deployment scenarios.

- *Generalization Across Network Types*: The proposed model's generalizability across different types of sensor networks deserves attention. While the current study focuses on a specific sensor network setup, the model's performance may vary when applied to diverse network architectures. Further investigations across various sensor network configurations will contribute to a more comprehensive understanding of the model's capabilities.

- *Challenges in Large-Scale Implementation*:

   A. **Increased Training Time:** As the size of the dataset and the number of features grow, the training time for the SVR model may increase. Consideration should be given to distributed computing or parallelisation strategies to mitigate this challenge.

   B. **Memory Requirements:** Large-scale implementation may demand significant memory resources, especially when dealing with extensive datasets. Efficient memory management or distributed computing frameworks could be explored to address this concern.

   C. **ACO Scalability:** The scalability of the ACO algorithm could be influenced by the complexity of the optimisation problem and the chosen parameter values. Sensitivity analysis and fine-tuning may be required for large-scale scenarios.

By transparently addressing these limitations, we aim to provide a balanced perspective on the proposed approach. These considerations highlight potential areas for future research and improvement, ensuring the continued refinement of the model for practical deployment in real-world intrusion detection and prevention scenarios.

## 6.2 Computational complexity analysis

1. **Time Complexity:**

The time complexity of the proposed intrusion detection and prevention approach primarily stems from two key components: the SVR model training and the ACO algorithm.

- *SVR Model Training*: The time complexity of training the SVR model is influenced by the number of training samples (n) and the number of features (m). With the adoption of efficient optimisation algorithms in popular machine learning libraries, such as scikit-learn, the SVR training process is generally linear or slightly super linear in the number of samples and features.

- *ACO Algorithm*: The ACO algorithm's time complexity is associated with the number of iterations (iterations) and the ant population (ants) size. Generally, ACO exhibits linear time complexity. However, the influence of parameters like the number of iterations and the size of the ant population needs consideration.

2. **Space Complexity:**

The memory requirements during the model training and optimisation processes determine the space complexity.

- *SVR Model*: The space complexity of the SVR model is primarily related to storing the model parameters. This complexity is generally linear in the number of features.

- ACO Algorithm: ACO's space complexity is influenced by the storage of pheromone matrices and solution constructions. It is also typically linear in terms of the number of features and the ant population size.

### 6.3 Real-world scenario examples and areas of application

1. *Urban Surveillance Networks*: In urban environments, WSNs are employed for surveillance to ensure public safety. The proposed intrusion detection and prevention approach can be instrumental in identifying anomalous activities, such as unauthorised access to secured areas or unusual movement patterns. The model can effectively distinguish between normal and suspicious behaviour by leveraging data from various sensors, including motion detectors and environmental sensors [10, 28].

2. *Industrial IoT (IIoT) Applications*: In industrial settings where IoT devices are extensively used for process monitoring and control, ensuring the security of these systems is paramount. The proposed approach can be applied to detect intrusions in Industrial IoT (IIoT) networks, safeguarding critical infrastructure from unauthorised access and potential disruptions. The model's adaptability allows it to address specific security concerns prevalent in industrial environments [29, 30].

3. *Precision Agriculture*: WSNs play a pivotal role in modern agriculture for monitoring soil conditions, crop health, and environmental parameters. The proposed model can enhance the security of these networks by detecting and preventing unauthorised access or tampering with sensor nodes [31]. It ensures the integrity of data used for precision agriculture practices, preventing malicious interference that could impact decision-making processes [32].

4. *Smart Home Security*: The proposed approach can offer robust intrusion detection capabilities in the context of smart homes equipped with sensor networks for automation and security. By analysing patterns in sensor data from motion detectors, door/window sensors, and other relevant devices, the model can distinguish between normal household activities and potential security threats, providing homeowners with advanced threat detection and prevention [33].

5. *Environmental Monitoring in Remote Areas*: Deploying WSNs in remote environmental monitoring scenarios, such as wildlife conservation or ecological research, necessitates reliable intrusion detection mechanisms. The proposed approach can contribute to securing these networks against unauthorised access, ensuring the continuity of data collection, and minimising the risk of interference in sensitive ecological studies [34].

Those mentioned above are a few real-world applications, but the research scope is not limited to these.

### Author Contributions

**Conceptualization:** C. Kishor Kumar Reddy, Mohammed Shuaib, Shadab Alam.

**Formal analysis:** C. Kishor Kumar Reddy, Vijaya Sindhoori Kaza, P. R. Anisha, Mousa Mohammed Khubrani, Mohammed Shuaib, Shadab Alam, Sadaf Ahmad.

**Funding acquisition:** Mousa Mohammed Khubrani, Mohammed Shuaib, Shadab Alam.

**Methodology:** C. Kishor Kumar Reddy, Vijaya Sindhoori Kaza, P. R. Anisha, Mousa Mohammed Khubrani.

**Project administration:** Shadab Alam.

**Validation:** Mohammed Shuaib.

**Visualization:** Vijaya Sindhoori Kaza, P. R. Anisha, Mohammed Shuaib, Sadaf Ahmad.

**Writing – original draft:** C. Kishor Kumar Reddy, Vijaya Sindhoori Kaza, Shadab Alam.

**Writing – review & editing:** P. R. Anisha, Mousa Mohammed Khubrani, Mohammed Shuaib, Sadaf Ahmad.

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
