## [Decision Letter · Decision Letter 0]

17 Jan 2024

PONE-D-23-44218Optimising Barrier Placement for Intrusion Detection and Prevention in WSNsPLOS ONE

Dear Dr. Khubrani,

Thank you for submitting your manuscript to PLOS ONE. After careful consideration, we feel that it has merit but does not fully meet PLOS ONE’s publication criteria as it currently stands. Therefore, we invite you to submit a revised version of the manuscript that addresses the points raised during the review process.

We look forward to receiving your revised manuscript.

Kind regards,

Dr. Rahul Priyadarshi

Academic Editor

PLOS ONE

Journal Requirements:

“The authors extend their appreciation to the Deputyship for Research& Innovation, Ministry of Education in Saudi Arabia, for funding this research work through the project number ISP-2024”

“The authors extend their appreciation to the Deputyship for Research & Innovation, Ministry of Education in Saudi Arabia, for funding this research work through the project number ISP-2024”

4. We note that your Data Availability Statement is currently as follows: [All relevant data are within the manuscript.]

Additional Editor Comments (if provided):

Reviewers' comments:

Reviewer's Responses to Questions

**Comments to the Author**

1. Is the manuscript technically sound, and do the data support the conclusions?

Reviewer #1: Yes

Reviewer #2: Yes

2. Has the statistical analysis been performed appropriately and rigorously? 

Reviewer #1: Yes

Reviewer #2: Yes

3. Have the authors made all data underlying the findings in their manuscript fully available?

Reviewer #1: Yes

Reviewer #2: Yes

4. Is the manuscript presented in an intelligible fashion and written in standard English?

Reviewer #1: Yes

Reviewer #2: Yes

5. Review Comments to the Author

Reviewer #1: Clarification on Method Integration: The paper would benefit from a clearer description of how Support Vector Regression (SVR) models and Ant Colony Optimization (ACO) algorithm are seamlessly integrated. A more detailed explanation would help readers understand the synergies between these components in achieving effective intrusion detection and prevention.

Algorithmic Parameters and Sensitivity: Provide a detailed discussion on the parameters used in the Ant Colony Optimization (ACO) algorithm and how their variations may impact the results. A sensitivity analysis would enhance the paper's depth, offering insights into the robustness and adaptability of the proposed approach.

In-depth Feature Ranking Discussion: Elaborate further on the feature ranking process and the specific attributes that contribute to barrier count estimation. Providing more details on how certain features were prioritized will enhance the transparency and understanding of the predictive modeling aspect of the research.

Validation Techniques: Clearly outline the validation techniques employed to assess the reliability of the proposed model. This may include cross-validation strategies or additional experiments that substantiate the generalizability of the findings beyond the dataset used in the study.

Discussion on Model Limitations: Address potential limitations of the proposed approach, such as its sensitivity to specific network conditions or the scalability of the solution. Acknowledging these limitations will strengthen the paper by providing a more balanced perspective on the proposed method.

Practical Implications: Discuss the practical implications of implementing the proposed approach in real-world Wireless Sensor Network (WSN) environments. Considerations such as hardware requirements, computational complexity, and ease of deployment will add valuable insights for practitioners and researchers.

Comparison with State-of-the-Art Methods: Expand the performance comparison section to include a comparison with state-of-the-art intrusion detection and prevention methods in Wireless Sensor Networks. This will provide a broader context for evaluating the significance of the proposed approach.

Discussion on Model Interpretability: Address the interpretability of the SVR models and the ACO algorithm in the context of intrusion detection. Explain how the results from these models can be translated into actionable insights for network administrators.

Real-world Scenario Examples: Provide specific examples or scenarios where the proposed approach has practical relevance and utility in real-world intrusion detection and prevention. This will help readers visualize the applicability of the research in diverse settings.

Language and Clarity: Ensure that the language used throughout the paper is clear and concise. Pay particular attention to technical terms and methodologies, ensuring that they are explained in a manner accessible to a broad audience of researchers in the field.

Reviewer #2: Algorithm Parameter Settings: Provide detailed information on the specific parameter settings used in the Support Vector Regression (SVR) models and Ant Colony Optimization (ACO) algorithm. A more thorough discussion on parameter tuning would help in reproducing and validating the results.

Validation Techniques: Elaborate on the validation techniques employed to ensure the reliability and generalizability of the proposed model. Specify whether cross-validation or other validation methods were used, and discuss how the model performs on unseen data.

Optimization Convergence Criteria: For the Ant Colony Optimization (ACO) algorithm, discuss the convergence criteria applied. Highlight how convergence is determined and the implications of these criteria on the precision of barrier placement.

Statistical Significance: While substantial improvements in accuracy metrics are reported, provide statistical significance tests to validate these improvements. This would strengthen the robustness of the results and enhance the confidence in the reported performance enhancements.

Computational Complexity: Discuss the computational complexity of the proposed approach, particularly addressing any potential challenges related to large-scale implementation. Include information on time and space complexity for a comprehensive understanding of resource requirements.

6. PLOS authors have the option to publish the peer review history of their article (what does this mean?). If published, this will include your full peer review and any attached files.

Reviewer #1: No

Reviewer #2: No

---

## [Author Response · Author response to Decision Letter 0]

2 Feb 2024

All the reviewer comments have been addressed, and a separate file explaining the response to each review comment has also been uploaded.

---

## [Decision Letter · Decision Letter 1]

9 Feb 2024

Optimising Barrier Placement for Intrusion Detection and Prevention in WSNs

PONE-D-23-44218R1

Dear Dr. Khubrani,

We’re pleased to inform you that your manuscript has been judged scientifically suitable for publication and will be formally accepted for publication once it meets all outstanding technical requirements.

Kind regards,

Dr. Rahul Priyadarshi

Academic Editor

PLOS ONE

---

## [Editor Report · Acceptance letter]

20 Feb 2024

PONE-D-23-44218R1 

PLOS ONE

Dear Dr. Khubrani, 

I'm pleased to inform you that your manuscript has been deemed suitable for publication in PLOS ONE. Congratulations! Your manuscript is now being handed over to our production team.

Kind regards, 

on behalf of

Dr. Rahul Priyadarshi 

Academic Editor

PLOS ONE